



# The influence of the addition of a reactive low SOA yield VOC on the volatility of particles formed from photo-oxidation of anthropogenic – biogenic mixtures

Aristeidis Voliotis[1], Mao Du[1], Yu Wang[1], Yunqi Shao[1], Thomas J. Bannan[1], Michael Flynn[1], Spyros N. Pandis[2,3], Carl J. Percival[4], M. Rami Alfarra[1,5] and Gordon McFiggans[1]

[1]Centre for Atmospheric Science, Department of Earth and Environmental Science, School of Natural Sciences, The University of Manchester, Oxford Road, M13 9PL, Manchester, United Kingdom

[2]Department of Chemical Engineering, University of Patras, Patras, Greece

[3]Institute of Chemical Engineering Sciences, FORTH, Patras, Greece

[4]NASA Jet Propulsion Laboratory, California Institute of Technology, 4800 Oak Grove Drive, Pasadena, CA 91109, USA.

[5]National Centre for Atmospheric Science, Department of Earth and Environmental Science, School of Natural Sciences, The University of Manchester, Oxford Road, M13 9PL, Manchester, United Kingdom

*Correspondence to*: A. Voliotis (*aristeidis.voliotis@manchester.ac.uk*) and G.McFiggans (g.mcfiggans.manchester.ac.uk)

**Abstract.** In this study, we investigate the influence of isoprene on the volatility of secondary organic aerosol (SOA) formed during the photo-oxidation of mixtures of anthropogenic and biogenic precursors. The SOA particle volatility was quantified using two independent experimental techniques (use of a thermal denuder and FIGAERO-CIMS) in mixtures of α-pinene/isoprene, o-cresol/isoprene and α-pinene/o-cresol/isoprene. Single-precursor experiments at various initial concentrations and results from previous α-pinene/o-cresol experiments were used as reference. The oxidation of isoprene did not result in the formation of detectable SOA mass in single-precursor experiments, however isoprene-derived products appeared in mixed systems, likely due to the increase in the total absorptive mass. Addition of isoprene resulted in mixture-dependent influence on the SOA particle volatility. Isoprene made no major change to the volatility of α-pinene SOA particles though changes in the SOA particle composition were observed, with volatility predicted reasonably based on the additivity. Isoprene addition increased o-cresol SOA particle volatility by ~5/15% of the total mass/signal, respectively, indicating a potential to increase the overall volatility that cannot be predicted based on the additivity. The addition of isoprene to the α-pinene/o-cresol system (i.e., α-pinene/o-cresol/isoprene) resulted in slightly less-volatile particles than those measured in the α-pinene/o-cresol systems. The measured volatility in the α-pinene/o-cresol/isoprene system had ~6% higher LVOC mass/signal compared to that predicted assuming additivity with a correspondingly lower SVOC fraction. This





suggests that any effects that could increase the SOA volatility from the addition of isoprene are likely outweighed from the formation of lower volatility compounds in more complex anthropogenic-biogenic precursor mixtures. Detailed chemical composition measurements support the measured volatility distribution changes and showed an abundance of unique-to-the-mixture products appearing in all the mixed systems accounting for around 30−40% of the total particle phase signal. Our results demonstrate that the SOA particle volatility and its prediction can be affected by the interactions of the oxidised products in mixed precursor systems and further mechanistic understanding is required to improve their representation in chemical transport models.

## 1. Introduction

Aerosol particles are ubiquitous in the atmosphere. They influence the Earth's climate directly, through scattering and absorption of solar radiation as well as indirectly by altering cloud properties (Ramanathan et al., 2001;Andreae and Rosenfeld, 2008). Moreover, particulate matter (PM) affects human health and is associated with millions of deaths and reduced life-expectancy (Cohen et al., 2017;Brunekreef and Holgate, 2002;Cohen et al., 2005). An accurate representation of the aerosol particles and their associated impacts in predictive models can contribute to the development of more effective mitigation policies.

The organic aerosol (OA) fraction comprises 20-90% of the total fine PM mass and may originate from numerous sources and chemical processes (Jimenez et al., 2009;Kanakidou et al., 2005). Secondary organic aerosol (SOA) is formed from the oxidation of volatile (VOCs), intermediate volatility (IVOCs) and semi-volatile (SVOCs) organic compounds in the atmosphere and accounts for a substantial part of the OA (Srivastava et al., 2018;Hallquist et al., 2009). The diversity of emission sources and transformation pathways of SOA results in a highly chemically complex mixture (Goldstein and Galbally, 2007) which is extremely challenging to represent accurately in predictive models (Shrivastava et al., 2017;Hallquist et al., 2009).

The majority of predictive models currently represent SOA in a simplified way (Tsigaridis et al., 2014;Shrivastava et al., 2017). The SOA particle mass is usually predicted based on the extrapolation of the SOA yields from single SOA precursor laboratory studies (Kanakidou et al., 2005). Recently, the volatility basis set (VBS; Donahue et al., 2006;Donahue et al., 2012) under the assumption of equilibrium absorptive partitioning has been developed and implemented in such models to predict the formation and evolution of SOA (Shrivastava et al., 2011;Giani et al., 2019;Koo et al., 2014;Tsimpidi et al., 2018). The SOA volatility distribution used in such models is obtained directly or indirectly from laboratory/field studies (Zhao et al., 2016;Giani et al., 2019). Additivity is implicitly assumed, as the co-existence of multiple reacting VOCs does not affect the volatility distribution of the SOA formed from each precursor. However, despite these advances, computer models still face challenges to accurately predict the SOA loadings (Shrivastava et al., 2017), potentially due to the lack of





parameterisations capturing explicit physical and chemical processes occurring as the OA forms and evolves in the atmosphere (Shrivastava et al., 2015;McFiggans et al., 2019;Jimenez et al., 2009;Donahue et al., 2012).

Recent laboratory evidence demonstrated that the SOA formation potential in the presence of multiple precursors can be affected by the mechanistic interactions between the products (McFiggans et al., 2019;Shilling et al., 2019;Berndt et al., 2018). The SOA particle volatility can also change due to those interactions, for example in the anthropogenic-biogenic

VOC systems investigated by Voliotis et al. (2021). It is important to understand the molecular interactions that have a central impact on the SOA formation and properties in the presence of multiple VOC precursors. Considering that the ambient atmosphere is a complex mixture of various anthropogenic and biogenic VOCs (aVOCs and bVOCs, respectively), these findings highlight the need to investigate the volatility changes in mixtures and their implications for the SOA formation and transformation.

The volatility of the oxidation products formed either in the gas or in the particle phase is largely controlled by the attained functionalities (McFiggans et al., 2010;Barley et al., 2009;Topping et al., 2011), as well as the molecular weight of the molecules (Kroll and Seinfeld, 2008;Shrivastava et al., 2017). The oxidation products of multiple coexisting VOC precursors may react with each other in either phase through radical termination reactions (Schervish and Donahue, 2020), leading to the formation of unique products. These interactions can also inhibit the formation of products that would have been

otherwise formed in simpler systems, a process known as product scavenging (McFiggans et al., 2019). In our recent work on mixtures of α-pinene and o-cresol, we have established a methodology to investigate the links between the SOA composition and volatility in experiments conducted in an atmospheric simulation chamber (Voliotis et al., 2021). More specifically, we showed that the unique products found in the mixture had high oxygen content, molecular weight (MW) and oxidation state ($\overline{(OSc)}$) resulting in lowering of the average particle volatility. At the same time, the scavenging of high MW

and low volatility α-pinene-derived products was increasing the SOA volatility. These processes resulted in a volatility distribution of the mixture being somewhere in-between those measured in the single-precursor experiments. Understanding the impacts of those interactions on the SOA particle volatility in multiple mixed precursor systems might therefore improve their representation in predictive models.

In this paper, we extend our previous work in mixtures of α-pinene and o-cresol by adding isoprene, the most abundant

bVOC with low SOA particle yield (Carlton et al., 2009) to the reacting mixtures. We investigate the SOA particle volatility changes in the photo-oxidation of isoprene-containing binary systems (α-pinene/isoprene and o-cresol/isoprene), and also ternary mixtures (i.e., α-pinene/o-cresol/isoprene). We then evaluate whether the photochemical evolution of mixed VOC systems leads to particles with comparable volatility to the sum of the individual systems assuming additivity. Detailed SOA chemical composition measurements are used to contextualise our findings and interpret any similarities or discrepancies

between the measured and the predicted SOA particle volatility.



## 2. Materials and methods

### 2.1. Experimental procedure

In this work, we assess the influence of isoprene on the volatility of SOA formed from the photo-oxidation of $\alpha$-pinene, $o$-cresol and their mixture in the presence of $NO_x$ and neutral inorganic seed particles using the Manchester Aerosol Chamber
(MAC). Three types of experiments were conducted: (a) single-precursor experiments, in which the photo-oxidation of each precursor was studied individually, (b) binary experiments, in which the photo-oxidation of mixtures of $\alpha$-pinene/isoprene and o-cresol/isoprene was studied and (c) ternary experiments in which all three precursors, $\alpha$-pinene, $o$-cresol and isoprene were oxidised simultaneously. Following Voliotis et al. (2021) and Voliotis et al. (2022) all experiments were designed to start with initial iso-reactivity; the injected VOC concentrations were selected based on the ratios of the inverse of their
IUPAC-recommended rate constants at 298K towards the OH (see Voliotis et al., 2022). This approach ensures that at the beginning of each experiment, each VOC in the mixture will have equal chances of reacting with the dominant oxidant and thereby they will have equal contributions to the first generation of products. The production of ozone during the experiments due to the presence of $NO_x$ (Atkinson, 2000), and the formation of oxidation products with their own reactivity towards both major oxidants, ensures that the systems will deviate from iso-reactivity to a greater or lesser extent during the
experiments.

In all single-precursor experiments, isoprene did not form SOA mass above the background levels (i.e. ~1 μg m$^{-3}$). Therefore, the isoprene single-precursor experiments are not discussed further. This effectively zero SOA particle mass yield of isoprene enables straightforward comparison of the composition and volatility between systems. In the binary systems, comparing the SOA composition and volatility of the mixture with those obtained at the respective single-precursor
experiments (e.g., $\alpha$-pinene/isoprene vs. $\alpha$-pinene) can provide direct evidence of the changes due the addition of isoprene. Similarly, in the ternary mixture, comparing its composition and volatility with the $\alpha$-pinene/$o$-cresol system can illustrate the influence of isoprene on the behaviour of that mixture. What is less straightforward, is the effect of the initial VOC concentrations used in the single, binary and ternary experiments. The use of different concentrations was necessary to achieve similar initial reactivity between the systems. Consequently, to investigate the effect of the initial VOC
concentration on the SOA particle composition and volatility, we conducted additional single-precursor experiments at half and one-third initial concentration for $\alpha$-pinene and at half the initial levels for $o$-cresol. The $\alpha$-pinene and $o$-cresol single-precursor at full reactivity and the $\alpha$-pinene/$o$-cresol mixture experiments have been described in detail in (Voliotis et al., 2021). Their results are summarised in the section comparing the composition and volatility across systems and reactivity levels.

In order to explore the role of mechanistic interactions in the mixtures, we estimate the expected SOA volatility in all the binary and the ternary mixtures assuming additivity. Details are provided in Section 2.4.3. Briefly, in the isoprene-containing binary systems, i.e., $\alpha$-pinene/isoprene and $o$-cresol/isoprene, the expected SOA particle volatility will be equal to



that of the $\alpha$-pinene and $o$-cresol SOA in the single-precursor experiments, respectively, adjusted to the same total absorptive mass. However, in systems where more than one precursor contributes to the SOA mass (i.e., $\alpha$-pinene/$o$-cresol and ternary) predictions may provide insight into the roles that mixing might play in SOA formation. The difference between the measured and the expected (predicted) volatility distributions can serve as a quantitative metric of the magnitude of the effect of the molecular interactions on the SOA volatility.

All experiments were performed under moderate relative humidity (50±5%) and room temperature (24±2 °C) in the presence of $NO_2$ (VOC/$NO_2$ = 6.2±1.4 ppb/ppb) and ammonium sulfate seed particles (54±11 µg m$^{-3}$). A summary of the initial experimental conditions is provided in Table 1. Each experiment was performed following procedures described in detail in Voliotis et al. (2021). Briefly, the "chamber stabilisation" phase (~1 h) was initialised when the chamber was filled with clean air in the absence of any of the reactants, followed by the "dark unreactive" phase (~1 h) where all the reactants (VOC, $NO_2$ and seeds) were added in the dark. The "experiment" phase (~6 h) started when the lights were turned on, indicating the initialisation of the photo-oxidation and SOA production. The data collected during the "chamber stabilisation" and the "dark unreactive" phase were used as baselines to correct the data obtained during the "experiment" phase (see Section 2.4.1). Prior and after each experiment the chamber was flushed with high flow rate (~3 m$^3$ min$^{-1}$) of purified air. The initial concentrations in the clean chamber were: particle number concentrations ≤15 cm$^{-3}$, $O_3$ ≤1 ppb and $NO_x$ ≤8 ppb (NO ≤6 ppb; $NO_2$ ≤2 ppb).

## 2.2. Manchester Aerosol Chamber

All the experiments were conducted at the 18 m$^3$ fluorinated ethylene propylene (FEP) MAC, operating as a batch reactor (Shao et al., 2022). The MAC is enclosed in a temperature and relative humidity-controlled housing. Due to its design, the chamber can expand and collapse as the air volume changes and thereby does not require dilution of the sample. The mixing and temperature control is achieved with the usage of an air conditioning unit (Denco C3-05), which circulates air around the chamber housing at high flow rate, setting its walls in constant motion and resulting in effective mixing throughout the experiments. A combination of two 6 kW Xenon arc lamps (XBO 6000 W/HSLA OFR, Osram) equipped with quartz glass filters and a series of halogen lamps (Solux 50 W/4700 K, Solux MR16, USA) is used to illuminate the chamber mimicking the atmospheric radiation spectrum as closely as possible (Shao et al., 2022). Steady-state actinometry experiments showed a photolysis rate of $NO_2$ ($J_{NO2}$) at 0.135 ± 0.024 min$^{-1}$, while the OH concentrations were estimated at around $1\times10^6$ molecules cm$^{-3}$. VOCs are produced by evaporating the corresponding liquids ($\alpha$-pinene, isoprene and $o$-cresol; Sigma Aldrich, GC grade ≥99.99% purity) in a gently heated glass bulb and are transferred to the MAC using an electronic capture device-grade nitrogen stream (ECD $N_2$; N4.8 purity ≥99.998%). An aerosol generator (Topaz model ATM 230) was employed to produce seed particles using aqueous solutions of ammonium sulfate of concentration 10 g L$^{-1}$ (Puratonic, 99.999% purity). $NO_2$ is





added to the chamber using custom-made cylinders (10% v/v) using ECD $N_2$ as carrier gas. Seeds, $NO_2$ and VOCs are added

155 to the chamber at high flow rate (~3 $m^3$ $min^{-1}$), ensuring rapid initial mixing.

**Table 1:** Summary of initial experimental conditions.

| Exp. no. | VOC | $NO_x$ (ppb) | VOC (ppb) | $VOC/NO_x$ (ppb/ppb) | Seed ($\mu g$ $m^{-3}$) | $SOA_{max}$ ($\mu g$ $m^{-3}$) | SOA Yield |
|---|---|---|---|---|---|---|---|
| 1 | α-pinene | 40 | 309 | 7.7 | 72.6 | 273.2 | 0.32 |
| 2 | α-pinene | 43 | 309 | 7.2 | 67.6 | 283.1 | n.a. |
| 3 | α-pinene | 26 | 155 | 6.0 | 45.7 | 68.6 | 0.21 |
| 4 | α-pinene | 18 | 103 | 5.7 | 51.0 | 31.5 | 0.15 |
| 5 | o-cresol | 98 | 400 | 4.1 | 50.8 | 28.2 | 0.15 |
| 6 | o-cresol | 56 | 400 | 7.1 | 67.6 | 23.5 | 0.13 |
| 7 | o-cresol | 71 | 400 | 5.6 | 36.0 | n.a. | n.a. |
| 8 | o-cresol | 40 | 200 | 5.0 | 51.3 | 22.8 | 0.11 |
| 9 | isoprene | 24 | 164 | 6.8 | 64.1 | 0.0 | 0 |
| 10 | isorpene | 23 | 164 | 7.1 | n.a. | 0.0 | 0 |
| 11 | isoprene | 14 | 55 | 3.9 | 42.2 | 0.0 | 0 |
| 12 | o-cresol/isoprene | 34 | 200/82 | 8.3 | 49.6 | 11.2 | 0.06 |
| 13 | o-cresol/isoprene | n.a. | 200/82 | n.a. | 57.0 | 9.4 | 0.05 |
| 14 | α-pinene/o-cresol | 52 | 155/200 | 6.8 | 48.3 | 122.3 | 0.26 |
| 15 | α-pinene/o-cresol | 30 | 155/200 | 11.8 | 57.0 | 131.4 | n.a. |
| 16 | α-pinene/isoprene | 33 | 155/82 | 7.2 | 63.7 | 96.6 | 0.17 |
| 17 | α-pinene/isoprene | 39 | 155/82 | 6.1 | 62.0 | 100.9 | 0.19 |
| 18 | α-pinene/o-cresol/isoprene | 60 | 103/133/55 | 4.9 | 66.1 | 33.3 | n.a. |
| 19 | α-pinene/o-cresol/isoprene | 78 | 103/133/55 | 3.7 | n.a. | 58.0 | 0.12 |

## 2.3. Instrumentation

160 Particle size distributions in the range of 10-670 nm were measured employing a scanning mobility particle sizer (SMPS; TSI DMA 3081 and CPC 3776, TSI Inc., USA). Non-refractory aerosol chemical composition was measured using a high-resolution aerosol mass spectrometer (HR-AMS), regularly calibrated according to well-known and previously published procedures (Jimenez et al., 2003;Allan et al., 2004;Voliotis et al., 2021). A thermal denuder (TD) installed upstream of the SMPS and HR-AMS and operated at 30-90 ºC in 12 steps was used to study the aerosol evaporation during the last two



hours of each experiment. During that period, the SMPS and HR-AMS sampling was alternated between the TD and the main sampling line (i.e., bypass) every 6 and 4 min, respectively. The Filter Inlet for Gas and Aerosols (Lopez-Hilfiker et al., 2014) coupled to an Iodide High-Resolution Time-of-Flight Chemical Ionisation Mass Spectrometer (Lee et al., 2014; hereafter FIGAERO-CIMS) was used to provide near real time gas and particle phase compositions. A semi-continuous gas chromatograph (6850 Agilent) coupled to mass spectrometer (5975C Agilent; hereafter GC-MS) equipped with a thermal desorption unit (Markes TT-24/7) was used to monitor the concentration of VOC precursors during the experiments. All the instruments were sampling air from the middle of the chamber using stainless steel and PTFE tubing (OD: ¼") for the particle and gas phase measurements, respectively.

The FIGAERO-CIMS was configured to operate in cyclic mode using the following procedure: i) 30 min of gas phase sampling at 1 L min$^{-1}$ and simultaneous particle collection on a PTFE filter (Zefluor, 2.0 µm pore size) at 1 L min$^{-1}$, ii) 15 min thermal desorption of the collected particles from ambient to 200 ℃ using ultra-high purity (UHP) $N_2$ as carrier gas, iii) 10 min soaking period, where the temperature was held constant at 200 ℃ and iv) 10 min of cooling period. The sample flow over the PTFE filter and at the exhaust of the ion molecule region (IMR) was continuously monitored using two MKS mass flow meters to ensure known volumes of air sampled. The instrument was run in negative-ion mode with I$^-$ reagent ion in all the experiments. The reagent ions were produced by passing $CH_3I$ and UHP $N_2$ over a $^{210}$Po radioactive source introduced directly into the IMR. Due to the particularly low organic concentrations in the first hours of the experiments with the *o*-cresol and *o*-cresol/isoprene systems as well as for comparison with the TD measurements, all the results shown here by the FIGAERO-CIMS correspond to the data obtained during the last FIGAERO-CIMS cycle of each experiment (i.e., > 5 h after lights on).

As a result of a lack of calibration standards and the experimental limitations associated with the FIGAERO-CIMS operation, the quantification of the observed signals remains challenging (Riva et al., 2019). Consequently, in this study, uniform instrument sensitivity was assumed for all the detected products. Detailed description of the instrumentation and the experimental setup is provided in Voliotis et al. (2021).

## 2.4. Background corrections and data treatment

The mass fraction remaining (MFR) of the aerosol particles after passing through the TD unit at each temperature step was calculated as the ratio of the SOA mass (measured by the HR-AMS) passing through the TD to the average SOA mass at the bypass line before and after each temperature step. The particle losses in each temperature in the TD were estimated based on characterisation experiments conducted with the same instrument configuration using sodium chloride particles and an SMPS measuring the total particle mass. The obtained SOA MFR were then corrected to account for particle losses in the TD (Voliotis et al., 2021). The HR-AMS collection efficiency was assumed to be the same at all temperatures used.





The Tofware package in Igor Pro (v. 3.2.1., Wavemetrics) (Stark et al., 2015) was employed for the data analysis and peak identification of the FIGAERO-CIMS data. $I^-$, $CH_2O_2I^-$, $I_2^-$ and $I_3^-$ were used for the mass-to-charge (m/z) calibration, resulting in an error less than 3 ppm. The HR analysis and peak identification were performed in the 190-550 m/z range based on the mass defect, until no possible formulas were available within a reasonable fitting error (<6 ppm). In all experiments conducted, this resulted in ≥70% of the total signal (excluding reagent ions) being assigned. In this work, only the assigned fraction of the signal is considered. The data collected in each thermal desorption (hereafter particle phase) and gas phase sampling cycles were grouped separately and each HR peak was background-corrected according to Voliotis et al. (2021). Briefly, in order to account for any potential instrument contamination in either phase, as well as any chamber background and/or filter/seed effects, a two-step correction procedure was followed. In the first step, the data collected during the assumed "instrument background" periods in each cycle were subtracted from the measured data to account of any potential instrument contamination. During the gas phase sampling, the instrument was periodically flushed with high purity $N_2$ in order to establish a dynamic gas phase "instrument background" signal, which was subtracted from the measurements. A corresponding "instrument background" was assumed for the particle phase measurements to be the 60th-90th second of the desorption cycle as shown in Voliotis et al., (2021). In the second step, the data collected during the "chamber stabilisation" phase (see Section 2.1) were subtracted from the gas-phase measurements to account for any potential background gas phase species in the MAC. Correspondingly, the data collected during the "dark unreactive" phase (see Section 2.1) were subtracted from the particle phase measurements to account for any potential chamber background and/or seed effects.

## 2.5. SOA particle volatility estimation

The SOA particle volatility was estimated here based on both the TD and the FIGAERO-CIMS measurements. The SOA particle volatility from the TD measurements was estimated using the algorithm of Karnezi et al. (2014). The algorithm is based on the thermodynamic mass transfer model of Riipinen et al. (2010) and employs forward and inverse modelling combined with an error propagation and minimisation procedure to interpret the MFR data. Briefly, the model selects appropriate volatility distributions expressed in the VBS framework, enthalpies of vapourisation and mass accommodation coefficients to fit the MFR data. Any potential temperature gradients in the TD are neglected, while the size-distribution of the aerosol particles is considered to be monodisperse, having diameter equal to the mean volumetric diameter (defined by the SMPS measurements). Furthermore, in addition to MFR and the mean volumetric diameter, the model uses the characteristics of the TD, such as the length of the heating section (0.51 m) and the aerosol residence time (31 s), and the SOA particle characteristics, such as the density (1.4 g cm$^{-3}$) and the total absorptive mass concentration ($C_{tot}$). Here, the volatility distributions were estimated in the $10^{-3}$-$10^2$ µg m$^{-3}$ $C^*$region.





The SOA particle volatility from the FIGAERO-CIMS measurements was estimated based on the gas and particle phase concentrations (ions m$^{-3}$) and absorptive partitioning theory (Pankow, 1994). Briefly, the partitioning coefficient ($f_p$) of each identified HR peak was calculated as:

$$f_{p,i} = \frac{Particle_i}{Particle_i + Gas_i}$$  (1)

where, $Particle_i$ and $Gas_i$ are the concentrations of each HR peak in the particle and gas phase (ions m$^{-3}$), respectively. Based on absorptive partitioning theory, the $f_p$ was converted to effective saturation concentration ($C^*$) using the total absorptive mass. For ease of interpretation, in the current manuscript we consider only the mass of organic material in the particle as being an effective absorptive medium in the partitioning of organic species. This is a reasonable approach since the nominally injected mass of inorganic seed particles was constant in all experiments and it is likely that the strong non-ideality in the mixing with inorganic components will lead to modest, if any, enhancement in component partitioning. The resultant volatility of the identified compounds in each system was expressed in logarithmically spaced bins in the VBS framework as:

$$C_i^* = \left(\frac{1}{f_{p,i}} - 1\right) C_{tot}$$  (2)

In either of these approaches, the sum of the volatility bins having $C^*<1$ µg m$^{-3}$ was considered as low volatile organic compounds (LVOCs), the sum those between $C^*=1-10^2$ µg m$^{-3}$ were considered as semi-volatile organic compounds (SVOCs) and the sum of those with $C^*>10^2$ were considered as intermediate volatile organic compounds (IVOCs). It should be noted that in both volatility estimation approaches, the bin with $C^*=10^{-3}$ may also include any extremely low volatility VOCs (ELVOCs) present. Similarly, in the TD results, the bin with $C^*=10^2$ may include compounds with even higher volatility that belong in the IVOC fraction.

## 2.6. SOA particle volatility prediction assuming additivity

The SOA particle volatility of the *α*-pinene/*o*-cresol and *α*-pinene/*o*-cresol/isoprene systems was estimated based on additivity, that is assuming no chemical interactions between the products of the various precursors. More specifically, the predicted mass fraction (PMF) of the products of each individual precursor (VOC$_i$) in the organic aerosol phase was estimated based on the VOC consumption ($\Delta VOC_i$) and the SOA yields measured ($Y_{VOCi}$) in the single-precursor experiments as:

$$PMF_{VOC_i} = \left(\frac{Y_{VOC_i} \times \Delta VOC_i}{\sum Y_{VOC_i} \times \Delta VOC_i}\right)$$  (3)



Subsequently, the volatility distribution of the SOA products of precursor $i$ in the binary/ternary systems were estimated by
weighting the sum of the volatility distributions measured in the single-precursor experiments with their predicted mass
fractions:

$$PC_j^* = \sum_j \left( MF_{VOC_i} \times C_{VOC_{i,j}}^* \right) \tag{4}$$

where, $PC_j^*$ is the predicted effective saturation concentration in the volatility bin $j$ and $C_{VOC_{i,j}}^*$ is the effective saturation
concentration of the products measured in the single-precursor experiments of each $VOC_i$ involved at the bin $j$.

The different total SOA concentrations ($C_{tot}$) in the experiments conducted for each system and reactivity level (Table 1) also
affect the gas/particle partitioning of the species (Pankow, 1994), and thereby the resulting measured volatility distributions.
Therefore, in order to predict the volatility distributions with this approach assuming additivity, all the measured volatilities
have to be adjusted to the same $C_{tot}$. For the volatility distributions obtained by the TD, we can adjust the measured volatility
distributions by running simulations at the desired total absorptive mass concentration (see Section 2.5). The volatility
distribution measured by the FIGAERO-CIMS at a certain $C_{tot}$ can be adjusted to any desired $C_{tot}$ by calculating the
partitioning coefficient ($f_p$) of each of the identified species, $i$, at the measured and the desired $C_{tot}$:

$$f_i = \left( 1 + \frac{C_i^*}{C_{tot}} \right)^{-1} \tag{5}$$

where $C_i^*$ is the estimated effective saturation concentration of each individual species derived from the partitioning
approach at the $C_{tot}$ levels of each individual experiment (eq. 2). By multiplying the particle phase FIGAERO-CIMS signal
of each individual compound $i$, with the ratio of the $f_p$ at the desired to measured $C_{tot}$, we can estimate signal contribution of
each compound due to the change in its partitioning behaviour as a result of the different $C_{tot}$.

## 3. Results

### 3.1. Single VOC system volatility at various reactivity levels

Figure 1 shows the retrieved volatility distributions from the TD and from the FIGAERO-CIMS, along with the identified
compounds in the particle phase expressed in the two dimensional volatility basis set space (2D-VBS) for the $\alpha$-pinene
experiments conducted at half and one-third initial reactivity (see 2.1.).



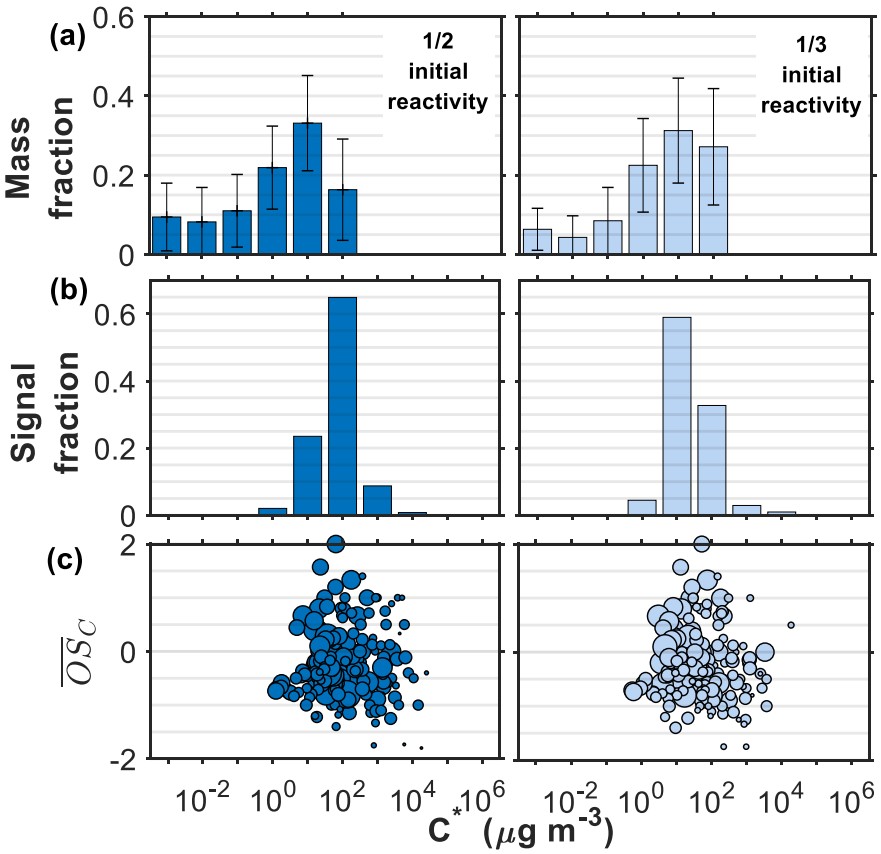

**Figure 1:** SOA particle volatility distributions from (a) the TD measurements (± retrieval uncertainty) and (b) from the
FIGAERO-CIMS measurements in the $\alpha$-pinene experiments conducted at half and one-third initial reactivity. (c) All
the products identified by the FIGAERO-CIMS are shown as a function of their volatility ($C^*$) and their average carbon
oxidation state ($\overline{OS_C}$) sized by the square root of their particle phase signal.

The volatility distributions obtained by the two independent methods in all experiments/systems were considerably different,
in line with previous studies (Stark et al., 2017;Voliotis et al., 2021). The signal-to-noise limitations of the FIGEARO-CIMS
can limit its ability to quantify compounds with low concentrations and low $C^*$. Similar problems can also appear for more
volatile compounds with low concentrations (Du et al., 2021). This often results in an artificial narrowing of the reported
volatility distributions. In spite of these limitations, the results from both approaches can be used to assess the volatility
changes when comparing such systems as discussed in detail in Voliotis et al., (2021).





The volatility distributions estimated in the α-pinene experiments at the two initial concentration levels appeared to be different according to the FIGAERO-CIMS. The estimated volatility distribution was shifted towards the lower volatility range as the SOA concentration was reduced from the full reactivity experiments (Fig. S1) to the one third (Fig. 1). This is

consistent with the predictions of the partitioning theory (Donahue et al., 2005). The uncertainty of the volatility distributions estimated from the TD technique did not allow the differentiation of the volatility of the two SOA systems in this case. The chemical composition of the products identified supports these trends and suggests that the lower the initial concentration, the stronger the contributions of products with $C^*= 10$ μg m$^{-3}$, $\overline{OSc}>0.5$ and nC $=7-10$. On the other hand, at the higher concentration experiments, compounds with $C^*=100$ μg m$^{-3}$ and $\overline{OSc}\leq0.5$ and nC$\geq8$ appear to contribute more to the total

signal (Fig. 1c and Fig. S1).

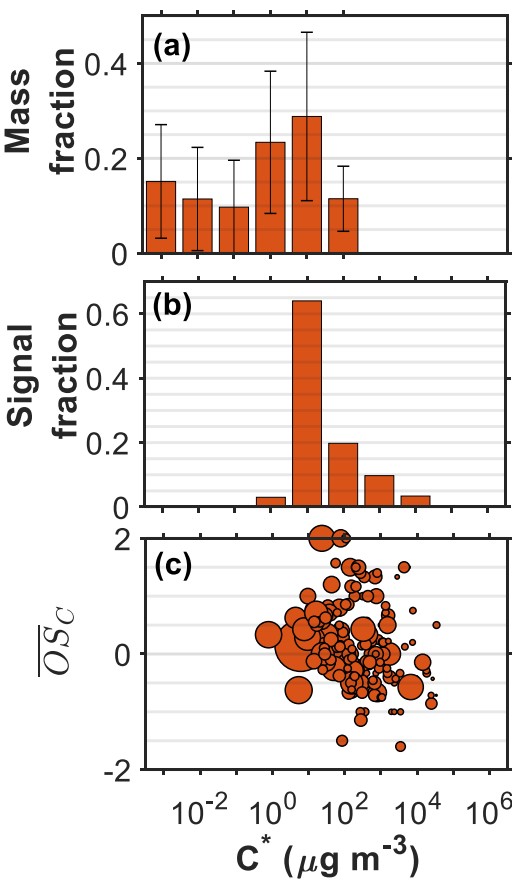

**Figure 2:** SOA particle volatility distributions from (a) the TD measurements (± retrieval uncertainty) and (b) from the FIGAERO-CIMS measurements in the *o*-cresol experiments conducted at half initial reactivity. (c) All the products that identified by the FIGERO-CIMS are shown as a function of their volatility ($C^*$) and their oxidation state ($\overline{OSc}$) sized by the

square root of their particle phase signal.


Figure 2 shows the corresponding volatility and 2-D VBS plots for the *o*-cresol experiment conducted at half reactivity. The volatility derived from the FIGAERO-CIMS had substantial contributions at the $C^*=10$ µg m$^{-3}$ volatility bin and around 40% of the signal in the $10^2$-$10^4$ µg m$^{-3}$ range. The distribution from the TD also peaked at $C^*=10$ µg m$^{-3}$ but suggested that 60% of the SOA had effective volatility less or equal to 1 µg m$^{-3}$. The SOA volatility distribution retrieved by the TD method was similar with that measured in the full reactivity experiments of Voliotis et al. (2021) shown in Fig. S1. According to the FIGAERO-CIMS measurements, a little higher contributions of more volatile material with $C^*>10$ µg m$^{-3}$ were observed at the lower initial reactivity experiment (40% versus 17% of the signal). The relatively narrow FIGAERO-CIMS volatility distributions in the *o*-cresol experiments are partially due to the dominance of the product(s) with elemental formula $C_7H_7NO_4$, which accounted for a substantial amount of the particle phase signal. For example, at the full initial reactivity experiments they accounted for 48% of the SOA signal (Voliotis et al., 2021). In the half reactivity experiment the $C_7H_7NO_4$ product(s) accounted for 34% of the total particle phase signal and leading to the lower fraction of the compounds with $C^*=10$ µg m$^{-3}$ and the corresponding increase of the more volatile fraction. The chemical characteristics of the compounds identified in these systems, expressed in the O:C vs H:C and the nC vs nO space showed somewhat higher signal contributions of more oxygenated compounds in the full initial reactivity experiments (Fig. S2), supporting these findings.

## 3.2. Volatility and composition of the mixtures

Figure 3 shows the measured and predicted volatility distributions from the TD and the FIGAERO-CIMS in all the systems investigated. The bars in the FIGAERO-CIMS volatility distributions show the contributions of the products derived from each individual precursor, those that had common elemental formulas between the systems as well as those that were only identified in each mixed system.

Although the SOA mass formed at the isoprene single-precursor experiments was found to be below our background (~1 µg m$^{-3}$), in all isoprene-containing systems studied here we were able to attribute a small fraction of the total FIGAERO-CIMS signal (≤6%) to isoprene-derived products. It is currently unclear whether the presence of these products is significant and/or this is could be attributed to a potentially high and/or differential sensitivity of our technique (e.g., Lee et al., 2014). Nonetheless, a likely explanation could be the significantly higher levels of total absorptive mass found in the binary systems opposed to the single-precursor isoprene would have favoured the partitioning of the more volatile species (Fig. S3). In support of this, it can be seen from Figures 4 and 5 that the majority of the isoprene-derived products have $C^*{\geq}100$ µg m$^{-3}$ which would cause these products to remain predominantly in the gas phase in the single-isoprene experiments where the total absorptive mass was ≤ 1 µg m$^{-3}$.



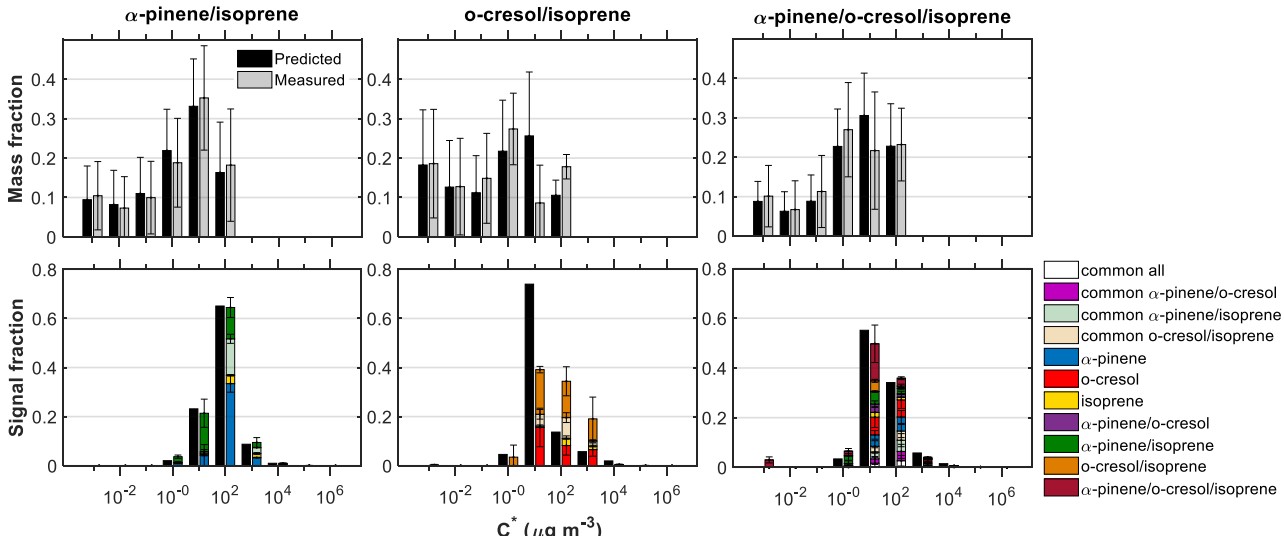

**Figure 3:** Measured and predicted based on the additivity SOA particle volatility distributions from the TD measurements (±retrieval uncertainty) (top panels) and from the FIGAERO-CIMS measurements (±1σ) (bottom panels) in all the systems investigated. The coloured bars in the top and bottom panels represent the measurements and the black bars the predictions. The coloured bars in bottom panels are separated to show the signal contributions of the products in each volatility bin. "Common" were classified as the products with common elemental formulae between the systems involved. The remaining products had elemental formulas that were unique to each single precursor or mixture system.

The SOA particle volatility measured in the $\alpha$-pinene/isoprene system from the TD method peaked at $C^*=10$ µg m$^{-3}$, while that from the FIGAERO-CIMS, showed highest signal contributions in the $C^*=100$ µg m$^{-3}$ bin. About 33% of the FIGAERO-CIMS total particle phase signal can be attributed to products that were uniquely identified in the mixture while about 42% and 6% due to products that have been identified in the $\alpha$-pinene and isoprene single-precursor experiments, respectively. The remaining ~18% is attributable to compounds with elemental formulas that have been observed in both single-precursor systems and the binary system ("common" compounds) and our technique is unable to resolve. In this system, the observed unique-to-the-mixture products (annotated as $\alpha$-pinene/isoprene in Fig. 3-5) had volatility spanning a wide range of 1 to $10^3$ µg m$^{-3}$, having highest contributions in the 10 and 100 µg m$^{-3}$ bins. Figure 4a depicts all the identified compounds in the O:C vs H:C, nC vs nO, nC vs $\overline{OSc}$ and $\overline{OSc}$ vs. $C^*$ spaces. The unique-to-the-mixture products with the highest signal contributions mostly had 7−10 carbon atoms, moderate oxygen content (3−5 oxygen atoms), relatively high $\overline{OSc}$ (>0.5) and H:C (>1.5). Considerable contributions are also observed for unique products with 6 carbon atoms or less, high $\overline{OSc}$ ($\overline{OSc}$>0.5) and O:C (O:C=~1). Examples from these unique products found in this binary system had assigned elemental





formulas: $C_6H_8NO_4$, $C_7H_{12}NO_5$, $C_8H_{14}NO_4$ and $C_9H_{12}O_3$. The $\alpha$-pinene-derived products in the mixture had similar chemical characteristics, though they exhibited higher signal contributions in the nC≥10 and $\overline{OSc}$=-1−0 ranges. The isoprene-derived products in the mixture had low nC (nC≤6) and relatively high $\overline{OSc}$ ($\overline{OSc}$>-1.8).

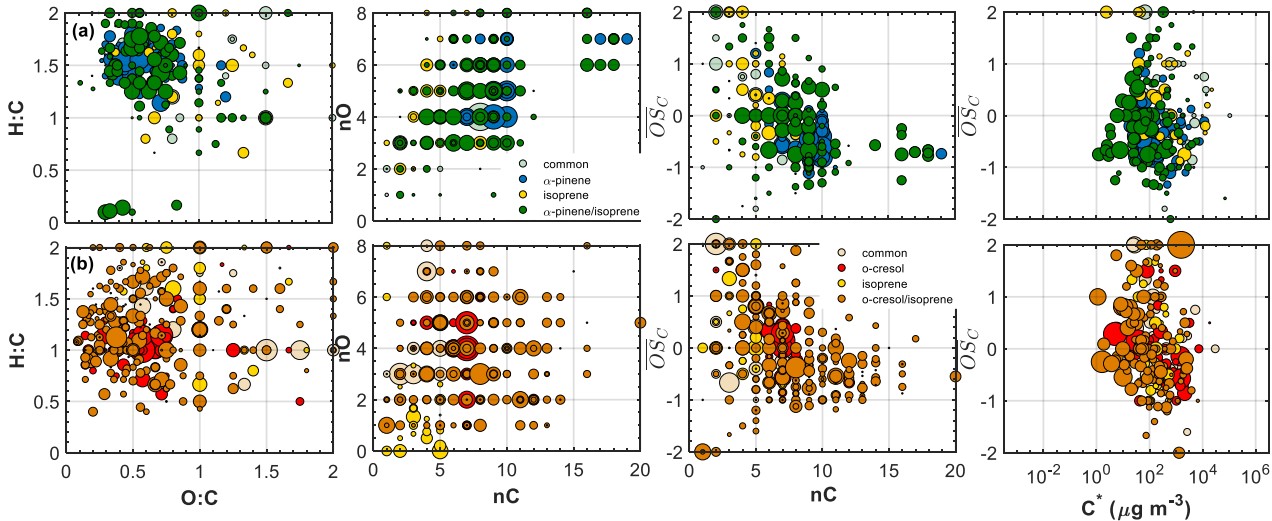


**Figure 4:** Hydrogen to carbon (H:C) by oxygen to carbon (O:C), number of carbon atoms (nC) by number of oxygen atoms (nO), carbon oxidation state ($\overline{OSc}$) by nC and $\overline{OSc}$ by effective saturation concentration ($C^*$) of all the individual products identified in the: (a) $\alpha$-pinene/isoprene and (b) $o$-cresol/isoprene systems. The products that were uniquely found in the mixture are depicted with different colours. All the symbols are sized based on the particle phase signal of each product in
the FIGAERO-CIMS.

The SOA volatility distribution measured in the $o$-cresol/isoprene showed high variability between the two experiments conducted (i.e., see error bars in Fig. 3 middle panels) even if they had similar SOA particle levels (Table 1). A relatively broad volatility distribution was observed both in the TD and FIGAERO-CIMS measurements. According to the TD
technique, there were substantial mass contributions both in the lower ($10^{-3}$ – $10^{-2}$ μg m$^{-3}$) and higher (10-100 μg m$^{-3}$) volatility bins. The estimated volatility distribution from the TD showed three peaks, at $10^{-3}$, 1 and 100 μg m$^{-3}$. On the other hand, according to the FIGAERO-CIMS measurements the SOA particles were considerably more volatile with a substantial fraction, around 20%, of the particle phase signal being in the IVOC region ($C^*$>$10^2$ μg m$^{-3}$). The SOA concentration in these experiments was of the order of 10 μg m$^{-3}$ so the presence of such high concentrations of IVOCs in the particulate
phase can be explained either by very high corresponding IVOC concentrations in the gas phase or by a significant underestimation of the less volatile SOA components by the FIGAERO-CIMS technique.





A large number of unique-to-the-mixture products were identified in the *o*-cresol/isoprene system that accounted for around 46% of the particle phase signal, having a wide range of volatilities and $\overline{OSc}$. The largest fraction of these products had appreciable signal contributions at high nC (>8) and O:C (≥0.7) and moderate $\overline{OSc}$ (~0.6). Examples of such products
identified in this system had assigned elemental formulas: $C_9H_8O_2$, $C_{11}H_{10}O_2$, $C_{10}H_{10}NO_4$, $C_{12}H_{11}O_4$ and $C_{13}H_{14}O_5$. The *o*-cresol derived products in the mixture on the other hand, were mostly centred at nC=7 and H:C=~1.1, at varying $\overline{OSc}$ and nO numbers, while the isoprene derived products had nC≤5 and H:C≥1.5 (Fig. 4b).

The volatility distribution retrieved from the TD measurements in the ternary system (i.e. α-pinene/o-cresol/isoprene)
indicated that around 70% of the SOA mass was approximately evenly distributed in the 1-100 μg m$^{-3}$ range with the rest of the material having lower volatility. The distributions derived from the FIGAERO-CIMS suggested that around 80% of the SOA signal was due to compounds with $C^*$ in the 10-100 μg m$^{-3}$, a result relatively consistent with that of the TD. According to the TD measurements, LVOCs represented 29% and SVOCs 71% of the SOA mass while according to the FIGAERO-CIMS there were 6% LVOCs, 87% SVOCs and 7% of the total signal were IVOCs.


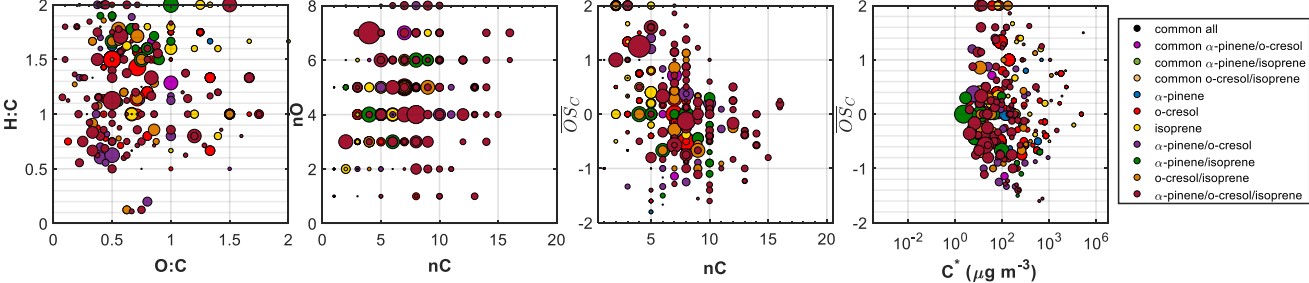

**Figure 5:** Hydrogen to carbon (H:C) by oxygen to carbon (O:C), number of carbon atoms (nC) by number of oxygen atoms (nO), carbon oxidation state ($\overline{OSc}$) by nC and $\overline{OSc}$ by effective saturation concentration ($C^*$) of all the individual products identified in the ternary system. The dots are sized based on the square root of their particle phase signal in the FIGAERO-
CIMS. The products that were common between all systems, those that were common between α-pinene, o-cresol and their mixture, those that were associated with either precursor, those that were unique products of each binary system as well as those that were unique products of the ternary mixture are depicted with separate colours.

Figure 5 shows all the identified compounds found in the ternary system in the O:C vs H:C, nC vs nO, nC vs $\overline{OSc}$ and $\overline{OSc}$
vs. $C^*$ spaces. In this ternary system, the SOA particle chemical composition was complex with 27% of the total particle phase signal of the FIGAERO-CIMS being attributed to products with common elemental formulas between experiments. Another 11% was attributed to α-pinene products 15% to the *o*-cresol products and 3% to isoprene products. The products that were found only in the binary systems accounted for 22% of the total signal; 6% for the unique products of the α-pinene/*o*-cresol, 5% for the *o*-cresol/isoprene and 11% for the unique products of α-pinene/isoprene system. Finally, 23% of
the signal was due to products found only in the ternary mixture. The majority of these unique ternary mixture products had





moderate volatility ($C^*$=1 μg m$^{-3}$), and relatively high number of carbons (more than 7). Their chemical characteristics, as illustrated in Figure 5, were spanning across a wide range of H:C, nO and $\overline{OSc}$. Their O:C was between 0.3 and 0.8. Characteristic examples of the elemental formula of such products were the: $C_8H_{12}NO_3$, $C_{10}H_{14}NO_5$, $C_{12}H_{10}O_4$, $C_{13}H_{15}NO_5$, $C_{14}H_{14}NO_3$ and $C_{14}H_{15}NO_5$.

## 4. Discussion

### 4.1. Influence of isoprene in binary mixtures

Considering that isoprene did not form any SOA mass above our background levels in all the single-precursor experiments using neutral ammonium sulfate seeds, the predicted mass contribution of its products in the binary mixtures is zero. Therefore, to investigate its influence to the SOA particle volatility in the binary mixtures under such conditions, we can predict the volatility distributions in all the isoprene-containing binary systems as being equal to that measured by the other precursor, adjusted at the same $C_{tot}$ levels. Considering that in the binary systems, only half of the total initial reactivity of the system was attributed to each of the precursors, the experiments at half initial reactivity were used to make such predictions.

In the α-pinene/isoprene mixture, the measured and predicted SOA volatility distribution estimated by both techniques were similar to the single precursor system (Fig. 1, 3 and S4), a result consistent with the almost identical total SOA levels (~60 μg m$^{-3}$; Fig. S3). In particular, the LVOC and SVOC mass fraction in the TD measurements were about 28 and 72%, while in the FIGAERO-CIMS measurements the LVOC, SVOC and IVOC content were about 0, 87 and 13%, respectively (Fig. S4), though caution should be employed in the interpretation of broad volatility classification since differences in the detail may be lost. This suggests that the addition of isoprene to the reacting α-pinene system had little effect on the volatility of the produced SOA that can be predicted assuming additivity. This may imply that any potential molecular interactions are unimportant in terms of SOA mass and/or that the unique-to-the-mixture products formed are contributing to the SOA mass appreciably but their similar volatility to the α-pinene products makes the net effect negligible. This is relatively consistent with the results of Ylisirniö et al. (2020) who reported minimal to modest changes in the SOA volatility from mixtures of real plant emissions and sesquiterpenes compared to single biogenic precursor experiments.

Even if the change in the overall SOA volatility distribution was small, the chemical information obtained in this system suggested changes in the SOA particle composition. More specifically the products that appeared in the nC=11−14, $\overline{OSc}$=−1−0.5 and $C^*$<100 μg m$^{-3}$ space in the α-pinene system were absent in the SOA particles of the binary system (Fig. 1 and Fig. 4). At the same time, unique-to-the-mixture products with nC≥14 and $\overline{OSc}$≥-0.75 and $C^*$<100 μg m$^{-3}$ were formed. Furthermore, unique-to-the-mixture products with moderate nC and similar chemical characteristics (i.e., H:C, O:C, $\overline{OSc}$) and volatility with those measured in the α-pinene single-precursor experiments were also formed (Fig. 4). These observations indicate that the mixing of isoprene with α-pinene can alter the chemical composition of the SOA particles



however, the net effect of these changes on the overall SOA volatility is minor. At least partially, this can be attributed to the similar elemental composition and volatility distribution that the unique-to-the-mixture products have with those found in the single-precursor experiments.

On the other hand, in the *o*-cresol/isoprene system, the volatility derived from both methods was significantly different to that obtained in the *o*-cresol system that resulted in different measured than predicted volatilities (Fig. 3). Particularly, the FIGAERO-CIMS measurements showed that the *o*-cresol/isoprene system had a ~10% increased measured than predicted IVOC fraction with a corresponding decrease in the SVOC fraction. The TD measurements showed a similar LVOC/SVOC split between the measured and predicted volatilites, yet the measured volatility distributions exhibited significantly higher

contributions in the $C^*{\geq}100$ µg m$^{-3}$bin (18% vs. 11% of the total mass; Fig. 3). Previous studies have showed that when isoprene is mixed with another aVOC, the SOA particle yields can be reduced (Jaoui et al., 2008;Vivanco et al., 2013;Ahlberg et al., 2017). This may imply that either the potential molecular interactions resulted in higher volatility products that remained in larger fractions in the gas-phase, or that isoprene was scavenging the available OH, thus reducing the SOA mass formed from the oxidation of *o*-cresol. In our set of experiments SOA particle mass yield of o-cresol was

unaffected by the presence of isoprene (Voliotis et al., 2022), suggesting that the former might be a more likely explanation. Alternatively, another explanation could be that the presence of isoprene could have altered the oxidation pathways of *o*-cresol, leading to the formation of higher volatility products or a combination of the above. Clearly, these results highlight the need for mechanistic studies investigating the SOA formation from that binary system.

The chemical composition of the products identified in the particle phase in each of the two systems (i.e., *o*-cresol and *o*-

cresol/isoprene) reveal significant differences. Almost all the *o*-cresol derived products observed in the single-precursor experiment in the nC=7 and $\overline{OSc}$>0.2 as well as those in the nC>7 and $\overline{OSc}$=-0.8−1.5 space, were not observed in the mixture (Fig. 4 and S2). At the same time, a number of unique-to-the-mixture products were identified in the nC>7 and $\overline{OSc}$>-1 and in the nC≤6 and at varying $\overline{OSc}$ spaces. Furthermore, isoprene-derived and unique-to-the-mixture products were also observed that had low nC and high $C^*$. The majority of these products exhibited relatively lower nC and O:C compared

to the products identified in the *o*-cresol system that likely supports the increase in the more volatile fraction of the mixture. These suggest, if anything, that the potential molecular interactions of the oxidised products in the gaseous or particulate phase in this system can increase the volatility of the mixture. Based on our results, the most likely explanation is the formation of unique-to-the-mixture products with different chemical characteristics and higher volatility and the simultaneous scavenging of *o*-cresol products with lower volatility (Fig. 4 and S2).




## 4.2. Influence of isoprene in ternary mixtures

To assess the influence of the effectively zero-yield isoprene to the SOA particle volatility of the $\alpha$-pinene/$o$-cresol mixture, both precursors with appreciable particle yields, the mass fraction of the SOA that corresponds to each precursor were estimated. Such estimations were achieved assuming additivity based on the corresponding reacted VOC concentrations
measured in the mixed system and the yields measured at the single-precursor experiments. Here, the high SOA yields of $\alpha$-pinene (Table 1) resulted in a high predicted mass fraction of the corresponding products in the SOA formed in the ternary system (~73±0.2%); so did in the $\alpha$-pinene/$o$-cresol system (69 ± 3%; See Fig. S5). These fractions were then used as weighting factors for the volatility distributions obtained in the single-precursor experiments in order to estimate the expected volatility distributions of the mixtures based on the additivity. Here, the volatility distributions of the $\alpha$-pinene at
one-third initial reactivity and the $o$-cresol at half reactivity were used to make such predictions.

The high SOA volatility observed in the $\alpha$-pinene single-precursor system from both techniques contributed substantially to the predicted volatilities in these systems that were generally higher than those measured (Fig. 1 and 3). The predicted volatility distributions from both techniques showed highest contributions at $C^*$=10 µg m$^{-3}$. In this system, the measured SVOC fraction was found to be slightly lower compared to that predicted both in the FIGAERO-CIMS (86 vs. 92% of the
total signal) and the TD measurements (71 vs, 77% of the total mass), with a corresponding increase in the LVOC fraction. Therefore, this suggests that the addition of isoprene has the potential to reduce the volatility of this mixture.

In more detail, the TD measurements showed that the ternary system had similar SOA volatility to the $\alpha$-pinene/$o$-cresol system (Fig. S4), yet with somewhat higher mass contributions in the $C^*\leq 1$ µg m$^{-3}$ region (55 vs. 49% of the total mass, respectively; Fig. 3 and S5). Similar trends are observed in the FIGAERO-CIMS measurements, where considerably higher
signal contributions in the $C^*\leq 10$ µg m$^{-3}$ region were observed in the ternary system compared to the $\alpha$-pinene/$o$-cresol (55 vs. 30% of the total signal, respectively; Fig. 3 and S5). This suggests, that the addition of isoprene in this system can affect the volatility of the SOA particles. Similarly to the other mixed systems, the $C_{tot}$ in the ternary mixture was lower than that measured in the $\alpha$-pinene/$o$-cresol system, roughly by a factor of 2.5 (Fig. S3). In this system, the adjusted volatility distributions of the $\alpha$-pinene/$o$-cresol system to the $C_{tot}$ levels of the ternary system from either method were almost identical
with those measured ($\leq$3% of the total mass/signal; Fig. S5 and S6). This demonstrates that any differences in the partitioning behaviour of the species due to the different $C_{tot}$ levels in the experiments conducted in these systems had a relatively small effect on the measured particle volatility distributions that are expressed in logarithmic scale. Consequently, the addition of isoprene in the $\alpha$-pinene/$o$-cresol system can further reduce the volatility of the system, a system that cannot also be predicted based on the additivity (see Fig. S5).

An alternative approach to assess the additivity in this complex system is to predict the SOA particle volatility based on those measured in a single and a binary system. For example, based on the $o$-cresol and the $\alpha$-pinene/isoprene systems. In





this case, the weights (or predicted mass fractions) can be calculated based on the VOC decay measured in the ternary system and the SOA particle yields of the o-cresol and the α-pinene/isoprene systems. This can be done for all the possible pairs of single and binary systems, i.e., *o*-cresol + (α-pinene/isoprene), α-pinene + (*o*-cresol/isoprene) and isoprene + (α-

pinene/*o*-cresol) and such predictions are shown on Fig. S7 from both techniques against the measured values. Interestingly, the measured volatility distributions are lower compared to those predicted in all cases and for both techniques, further confirming that the chemical interactions occurring in this system will lower the resultant SOA particle volatility. Clearly, however, the degree of the discrepancy between the measured and the predicted volatility distributions is dependent on the calculation method. This illustrates that any volatility prediction (and perhaps similarly, the SOA particle yield predictions),

will likely be unable to capture the complex interactions of the mixed systems. This should probably be expected considering that the $NO_x$ trends and the production of $O_3$ at similar initial $VOC/NO_x$ can substantially vary between single precursor or mixture experiments (Voliotis et al., 2022), suggesting the chemical regime of the experiments is system-dependent. Further, the differential reactivity of the oxidised products towards the OH and $O_3$ along with radical cross reactions that evidently lead to the formation of unique-to-the-mixture products will ensure deviation from the additivity to an unknown extent.

The chemical composition of the SOA formed in the ternary mixture was highly complex. The oxidation products of α-pinene and *o*-cresol contributed to the SOA considerably but nearly half of the FIGAERO-CIMS particle phase signal was attributed to unique-to-the-mixture products found in the binary systems and in the ternary mixture (Fig. 3, bottom-right panel). The elemental composition of all the unique products spanned a wide range in the O:C, H:C, nC and nO space, making the comparison between the systems challenging (Fig. 5). Overall, the largest fraction the unique-to-the-mixture

products (about 80%) appeared to have volatilities in the $C^*=1-10$ µg m$^{-3}$ range, suggesting that they were semi-volatile. It is unclear the extent that the identified unique-to-the-mixture products are due to changes of the oxidation pathways of each precursor involved due to the mixing or as radical cross reaction products or a combination of the above. Mechanistic studies and/or analytical studies that can provide molecular structure information might help identify the characteristics of those products and their formation pathways.

Previously we have shown that the unique products of the α-pinene/*o*-cresol system have, on average, higher molecular weight, average carbon oxidation state, oxygen content and consequently lower volatility than the products of either precursor (Voliotis et al., 2021). The simultaneous scavenging of higher molecular weight and lower volatility α-pinene products in that system resulted in a volatility distribution in-between those measured in the single-precursor systems. Here, we quantify that the molecular interactions of the products may lower the overall volatility of the α-pinene/*o*-cresol system

by ~6-11% (see Fig. S5). Expanding this to the ternary system with the addition of isoprene, we also find that the molecular interactions still leads to the formation of lower volatility SOA than that predicted based on the additivity, despite that the interactions from the mixing of *o*-cresol and isoprene likely lead to higher volatility SOA (Fig. 3).





The results shown above suggest that predicting the SOA particle volatility based on the additivity cannot always represent all the mixtures of anthropogenic and biogenic precursors. The molecular interactions that may occur between the oxidised products in mixtures have therefore the potential to alter the SOA composition and volatility. Clearly, mechanistic studies are needed to unravel the nature of those interactions and may help to develop advanced frameworks that can capture their trajectories.

### 4.3. Atmospheric implications

The atmospheric environment is a complex mixture of thousands of anthropogenic and biogenic VOCs, IVOCs and SVOCs reacting simultaneously. To date, the prediction of the SOA loadings in chemical transport models is carried out assuming that the SOA yields of one precursor for a given SOA concentration and VOC/NO$_x$ ratio are not affected by the presence of the other organic vapours (Tsimpidi et al., 2018). The results of this study show that upon mixing anthropogenic and biogenic precursors, the underlying molecular interactions have the potential to alter the SOA composition and volatility. The effect is less pronounced in the biogenic-biogenic mixtures investigated, but can be noteworthy in anthropogenic-biogenic mixtures. Previous ambient observations showed increased SOA particle contributions when anthropogenic and biogenic emissions were mixed (Hoyle et al., 2011;Shilling et al., 2013;Shrivastava et al., 2019). Our results show that the unique-to-the-mixture products of the anthropogenic-biogenic mixtures may have either higher (i.e., *o*-cresol/isoprene) or lower volatility (i.e., *α*-pinene/*o*-cresol), however in more complex mixtures (i.e., *α*-pinene/*o*-cresol/isoprene) the formation of lower volatility products likely outweighs the formation of products with higher volatility. Consequently, it may partly explain the increased SOA particle loadings when biogenic and anthropogenic emissions are mixed.

It should however be noted that the experiments conducted in this study were performed in a batch reactor using high concentrations of VOC precursors, in order to form adequate amounts of SOA particles. This is expected to promote the RO$_2$-RO$_2$ cross reactions and suppress the RO$_2$+HO$_2$ reactions (Peng and Jimenez, 2020;Schervish and Donahue, 2020) which are thought to be the main radical termination pathway in the atmosphere (Tan et al., 2018), in addition to RO$_2$+NO on certain occasions (Orlando and Tyndall, 2012). Therefore, the increased fraction of unique-to-the-mixture products observed in this study might be associated with the specific oxidative conditions defined by our experimental setup. Considering that the unique-to-the-mixture products were found to decrease the SOA particle volatility in the *α*-pinene/*o*-cresol and in the ternary systems while they increased the volatility in the *o*-cresol/isoprene system, it could be hypothesised that in low RO$_2$:HO$_2$ conditions (i.e., RO$_2$:HO$_2$<1) that effect could be supressed to an unknown extent. Although we are unable to quantify the mass fractions of the unique products due to analytical limitations, our results might still represent certain atmospheric conditions. Such an example might be clean environments where the RO$_2$:HO$_2$ has been found to be high (RO2:HO2>1; Stevens et al., 1997;Carslaw et al., 2002). Nonetheless, our scope here is not to provide volatility distributions that can be extrapolated globally, rather than to demonstrate that the molecular interactions have the potential to alter the SOA particle composition and volatility, with consequent implications for the air quality and climate predictions.



## 5. Conclusions


In this study, we extended our previous work in mixtures of α-pinene and o-cresol with the addition of a low yield precursor, isoprene, and we investigated the SOA particle volatility in binary mixtures of biogenic-biogenic (α-pinene/isoprene) and anthropogenic-biogenic (o-cresol/isoprene) precursors in a batch reactor. We further moved beyond the binary mixtures by combining the above precursors simultaneously in ternary mixtures (α-pinene/o-cresol/isoprene).


The α-pinene/isoprene system had comparable volatility distribution to that measured in the α-pinene single-precursor experiments that resulted to similar volatility predictions assuming additivity. Therefore, the SOA volatility from the oxidation of α-pinene was not influenced by the addition of isoprene. Nonetheless, there were evident changes in the SOA chemical composition with the suppression in the formation of some of the α-pinene products and the formation of products that were uniquely observed in the mixture. The unique products had similar volatility distribution to the components of the


α-pinene SOA system thus potentially explaining the small effect of their formation on the overall SOA volatility formed in the binary system.

In contrast, the addition of isoprene to o-cresol reacting system resulted in the increase of the contribution of higher volatility components to the SOA composition. Particularly, the measured volatility was higher than the one predicted by up to ~5-15% of the total mass/signal. The unique-to-the-mixture products contributed significantly to the total signal and a


significant fraction of which had lower number of carbon and oxygen atoms and higher H:C compared the products identified in the o-cresol experiments, supporting the observed higher volatility.

The measured SOA particle volatility of the ternary system was lower than that predicted by ~6% of the total mass/signal, demonstrating that the molecular interactions from the addition of isoprene in the α-pinene/o-cresol reacting system can reduce the overall volatility. The SOA chemical composition of that mixture was diverse with contributions of products from


all the precursors involved, unique products of all binary systems, as well as unique products of the ternary mixture, making the comparison between systems challenging.

The results described in this study suggest that the changes in the products formed when mixing anthropogenic and biogenic precursors might result in a net decrease or increase of the overall SOA volatility that is system-dependent. On the other hand, the similarity in the predicted and the measured volatility distributions in the mixed biogenic system (i.e., α-


pinene/isoprene) show that the changes in the products formed have a modest influence to the SOA volatility. Therefore, the molecular interactions that result from the mixing of anthropogenic and biogenic precursors can have a varying effect on the SOA volatility distribution of the systems and are not negligible. Clearly, the individual precursors involved and thereby the resultant oxidation products determine the magnitude of those interactions as well as the characteristics of the unique products formed, in turn affecting the volatility distribution of the SOA particles. The attribution of these changes to specific

processes remains unknown and highlights the need for mechanistic studies. The results presented here capture only certain atmospheric conditions, therefore the investigated chemical regimes, oxidants and VOCs should be extended.

**Data availability**

All the data used in this work are available upon request from the corresponding authors.

**Competing interests**

The authors declare that they have no conflict of interest.

**Author contributions**

AV and GM developed the concept of this paper. AV, YW, YS and MD conducted the experiments. SNP provided the volatility retrieval algorithm. TJB provided on-site help deploying the FIGAERO-CIMS. MF provided on-site help deploying the TD unit. AV conducted the data analysis and wrote the manuscript with inputs from all co-authors.

**Acknowledgements**

The Manchester Aerosol Chamber received funding from the European Union's Horizon 2020 research and innovation programme under grant agreement no. 730997, which supports the EUROCHAMP2020 research programme. AV acknowledges the Presidents Doctoral Scholarship from the University of Manchester and the support from the Natural Environment Research Council (NERC) EAO Doctoral Training Partnership. MRA acknowledges funding support from the
Natural Environment Research Council (NERC) through the UK National Centre for Atmospheric Science (NCAS). Instrumentational support was funded through the NERC Atmospheric Measurement and Observational Facility (AMOF). CJP work was carried out at Jet Propulsion Laboratory, California Institute of Technology, under contract with the National Aeronautics and Space Administration (NASA), and was supported by the Upper Atmosphere Research and Tropospheric Chemistry Programs.

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
