# Peer review of "The influence of the addition of a reactive low SOA yield VOC on the volatility of particles formed from photo-oxidation of anthropogenic – biogenic mixtures"

_Atmospheric Chemistry and Physics, 2022_

## Author Comment (AC1)

In this manuscript the authors report on experiments where they investigated the volatility of SOA particles in chamber studies, using both single precursor and mixed precursor experiments. The precursors were chosen to present two important biogenic VOC (isoprene and α-pinene), and a VOC to represent anthropogenic emissions (cresol). Their findings show that simple interpolation from single precursor data does not always yield correct predicted values for the mixed systems compared to actual measured values, bringing into question the validity of using such an approach in many models.

The overall results presented in the manuscript are good and an important addition to our understanding of the complexity of atmospheric SOA formation and one of the relevant properties of SOA, its volatility. There are few minor revisions (detailed below) and technical comments that would need addressing, but after those have been successfully handled, the work can be accepted for publication.

We'd like to thank Referee 2 for their positive comments and to respond to the general and detailed comments as follows (reviewer comments in black and our responses in blue; the line numbers referred throughout are referring to the original manuscript):

General comments:

1.  More discussion on the atmospherical relevance of the results is in order. The used [VOC] concentrations and seed concentrations are quite high, and may impact the behaviour of the system both chemically and physically. The effect of this in the experiments is pointed out in the manuscript lines 294-295, but should be further discussed later in the manuscript as well. Similarly, the [OH] reported is quite low in comparison to the [VOC], and it is much closer to atmospherically relevant concentrations. How does this affect the interpretation and relevance of the results? The possible change of reaction pathways is mentioned ($RO_2$ + $HO_2$ to $RO_2$+$RO_2$), but the effect and implications for the results could be discussed further.

Indeed, in our experiments the VOC concentrations were particularly high to ensure enough SOA particle mass formed for our subsequent offline filter analysis (see Voliotis et al., 2022). The seed concentrations were relatively high (~ 50 μg m$^{-3}$), though not unrealistic (see e.g., Cao et al., 2017), and necessary to provide a condensation medium to compete with the chamber walls and prevent nucleation. The oxidant levels (OH and $O_3$) were relatively low and indeed much closer to atmospherically-relevant levels.

As we discuss in the section 4.3 of the original manuscript and particularly in L546-559, the high VOC concentrations used in this study would mean a $RO_2$-$RO_2$ dominated chemical regime instead of a $RO_2$-$HO_2$ that might be more atmospherically relevant. This suggests that our experimental conditions might favour radical cross reactions. Therefore, perhaps the relatively high observed fraction of products that

were unique in mixtures (and likely derive from $RO_2$-$R'O_2$ termination reactions) can be partly attributed to these experimental conditions (see L550-551). So does any consequent effect of this fraction to the volatility, to a truly, unknown extent (see L552-554).

As can be seen from Voliotis et al., (2022), the VOC(s) were not fully consumed in any of our experiments, likely as a result of the low oxidant and high VOC(s) concentrations. This might suggest continuous generation of earlier generation products resulting in slower aging and likely higher volatility. This has been included in the discussion of the revised manuscript.

Nonetheless, as we state in the L557-560, it should be noted that our scope here is to demonstrate that the molecular interactions have the potential to alter the SOA particle composition and volatility rather than to provide volatility distributions that can be extrapolated globally.

2. It is mentioned that due to low organic mass produced in some experiments, the last FIGAERO sampling cycle has been chosen for each experiment for further analysis (lines 180-183). This is a good approach, however it would be better to include the OH exposure for different samples, and add discussion on the potential effect of (possibly) differing OH exposures.

As it is stated in the L180-183, we selected the last cycle of the FIGAERO primarily for comparison with the concurrent TD measurements that are reported here.

It should be noted that in our experiments we have a combined OH and $O_3$ exposure with the latter being formed secondary via the $NO_2$ photolysis as well as through as VOCs are consumed through the NO reaction with $HO_2$ (see L100-150). Therefore, in the systems containing unsaturated VOC(s) (or unsaturated oxidation products), $O_3$ will contribute to the oxidation of the VOC(s) and their products to an unknown degree, making the comparison of the systems' exposure challenging. The trajectories of oxidation between the systems and their effect on the SOA formation and evolution is thoroughly discussed in our companion paper; see Voliotis et al. (2022).

Nonetheless, it might worth mentioning that we have estimated the OH concentrations in all systems; in $o$-cresol-containing systems based on its decay rate and in non-$o$-cresol-containing via subtracting the expected VOC decay due to the $O_3$ concentration (details of the method are provided in Voliotis et al., 2022). The average OH concentration from all systems was estimated at $\sim 8 \pm 3 \times 10^5$ molec cm$^{-3}$, suggesting that the OH exposure between the different systems was comparable.

3. How were the FIGAERO-CIMS data calibrated? This should be made more clear, especially as C* can also be directly derived from measured FOGAERO-CIMS data, if there are proper $T_{max}$ calibrations done.

The FIGAERO-CIMS data were not calibrated in this study for C*. We have previously attempted to calibrate our FIGAERO-CIMS with the homologous series of polyethylene glycols (PEGs) that are recently being used as a $V_p$-$T_{max}$ calibration standard and we have contrasted their results with the partitioning calculations and TD-AMS measurements (see Voliotis et al., 2021). The volatility distributions obtained from the $T_{max}$ calibrations were unrealistically high with the vast majority of the signal being accumulated in C*>100 μg m$^{-3}$ at SOA particle mass loadings >200 μg m$^{-3}$, inconsistent with the absorptive partitioning theory. We attributed this to the method we selected to introduce the calibrants in the instrument (syringe method) that was more recently shown to have an effect on the $V_p$-$T_{max}$ relation (Ylisirniö et al., 2021), as well as to unquantified matrix effects during the desorption (Schobesberger et al., 2018). On the other hand, the volatility distributions obtained from the partitioning calculations were realistic and broadly comparable with the concurrent TD-AMS measurements. Importantly, despite the challenges in the volatility quantification from the FIGAERO-CIMS, we showed that our method is able to capture the volatility changes between the systems. Therefore, in this companion and follow-up paper, we have selected the partitioning method to illustrate the volatility changes and their effect in the volatility predictions as a result of the mixing of the precursors. The application of this method to our data is detailed in section 2.5 of the original manuscript.

Overall, we agree with the reviewer that some additional information might be needed for our reasoning in selecting the partitioning method therefore additional text has been added in the methods section of the revised manuscript.

Detailed comments:

4. lines 120-121 and 125: what is meant by additivity and predictions could be introduced here as well with a sentence.

A sentence was added in the revised manuscript.

5. lines 136-138: it would be useful to provide background [VOC] concentration as well. Especially as in the isoprene case the "background" mass formation was mentioned to be 1μg m$^{-3}$.

MAC is equipped with a VOC scrubber and our pre- and post-experiment procedures ensure VOC mixing ratios close or below our limit of detection (~0 ppb). This is shown in our characterisation paper (Shao et al., 2022) and mentioned explicitly in the revised manuscript. The 1μg m$^{-3}$ background organic particle mass is a result of the impurities from the seed generation procedures (in the order of

~1-3% of the total mass). Recently, Wu et al. (2022) have demonstrated the importance of this for atmospheric science studies and its implications remain largely unknown.

6. line 145: "setting its walls in constant motion" has this noticeable effect on particle wall losses? The data used in the study is wall loss corrected at least regarding particles?

The agitation of the chamber walls increases the losses of the particles but ensures effective mixing. The characterisation of the MAC showed the effect of the mixing to the losses of particles as a function of their size (see Fig. 5 in Shao et al., 2022). The data used for the SOA particle yield calculation were wall-loss corrected as detailed in Voliotis et al. (2022). For the partitioning calculations and volatility retrieval however, the uncorrected data were used to reflect actual amount of absorptive mass being present instead of the projected, wall-loss corrected.

7. line 153: "using aqueous solutions of ammonium sulfate of concentration 10 g L$^{-1}$." How does the size distribution of the seed population look like? To condensing vapours, total surface area is more relevant than total mass, how does the seed surface area compare between experiments?

We agree with the reviewer that the stock solution concentration might not be very insightful metric for the characteristics of the seed generated. The seed population generated had consistently a mode diameter of ~70 nm and a total surface area in the order of ~$10^9$ nm$^2$/cm$^3$. The average mode diameter and total surface area obtained during the "dark uncreactive" phase on were $67 \pm 6$ nm and $1.2 \pm 0.4 \times 10^9$ nm$^2$/cm$^3$. This information is now included in the revised manuscript.

8. line180-183: "Due to the particularly low organic concentrations in the first hours of the experiments with the o-cresol and o-cresol/isoprene systems as well as for comparison with the TD measurements, all the results shown here by the FIGAERO-CIMS correspond to the data obtained during the last FIGAERO-CIMS cycle of each experiment (i.e., > 5h after lights on)." how do the OH exposures between chosen samples compare with each other? Can they be directly compared in this manner? Depending on the results of the OH exposure calculation, this needs to be discussed further.

See our response to comment #2. Based on our estimation of the average OH, the OH exposure between systems was comparable. Our experiments however were conducted in the presence of both OH and O$_3$, both contributing to the oxidation of the parent VOC(s) and their products. As we comprehensively discuss in Voliotis et al., (2022), and also state in the L505-508, the chemical regime of the experiments appears to be system-dependent. This is a particularly interesting finding of our programme of study that strengthens our main conclusion; the complex interactions occurring in mixed VOC systems can lead to unpredictable changes in the SOA formation potential and properties.

9. lines 184-188: "As a result of a lack of calibration standards and the experimental limitations associated with the FIGAERO-CIMS operation, the quantification of the observed signals remains challenging (Riva et al., 2019). Consequently, in this study, uniform instrument sensitivity was assumed for all the detected products. Detailed description of the instrumentation and the experimental setup is provided in Voliotis et al. (2021)." What calibrations have been done? Gas phase and/or particle phase sensitivity calibrations? Voltage scanning? $T_{max}$/temperature ramp calibration? Due to the relevance of the FIGAERO-CIMS data and method for the results and their interpretation, it is important to include this information here as well.

We kindly refer the reviewer to the subsequent sections 2.4 and 2.5 of our original manuscript, where our background correction and data treatment procedures are detailed, as well as our response to the comment #3 above. Briefly, we did not conduct any compound-specific sensitivity calibrations, applied any voltage scanning or $T_{max}$-$V_p$ calibrations to the data shown in this paper. We did however filter out the ions that had signal-to-noise ratio below two, as described in Voliotis et al., (2021). This information is now included in the revised manuscript.

10. line 206-208: "During the gas phase sampling, the instrument was periodically flushed with high purity $N_2$ in order to establish a dynamic gas phase "instrument background" signal, which was subtracted from the measurements" Was the $N_2$ flow humidified? Was there a significant change in instrument response due to changing [$H_2O$] in the IMR?

At the time we conducted this series of experiments we did not have the capability to humidify our $N_2$ background flow. This consequently affects the instrument response between the sampling and background mode (Lee et al., 2014) and will lead to very small changes in the magnitude of the signal. Further, as we are only reporting here relative changes of the signal in the gas vs. particle mode between systems, we do not expect this to affect our conclusions.

11. lines 208-213: "A corresponding "instrument background" was assumed for the particle phase measurements to be the $60^{th}$-$90^{th}$ second of the desorption cycle as shown in Voliotis et al., (2021). In the second step, the data collected during the "chamber stabilisation" phase (see Section 2.1) were subtracted from the gas-phase measurements to account for any potential background gas phase species in the MAC. Correspondingly, the data collected during the "dark unreactive" phase (see Section 2.1) were subtracted from the particle phase measurements to account for any potential chamber background and/or seed effects." This is mostly very nice and thorough. However, were there no filter blanks taken during the experiments? There can be a measurable effect of gas phase collecting on filter, leading to seemingly higher signal for

certain ions, and this effect will not be captured by using the tail of the thermogram or samples taken before reactions start.

We have taken two blank filters in every experiment; one during the "chamber stabilisation" phase (chamber filled with clean air only) and another during the "dark unreactive" phase (chamber filled with the VOC precursor(s), NOx and seed particles). Here, we did not used the particle data collected at the "chamber stabilisation" phase; this cycle was serving as an enhanced filter cleaning cycle, ensuring low background on the filter. As it is stated in the L211-213, the particle phase data collected during the "dark unreactive" phase were used as blanks and were subtracted from the measurements to account for any background signals and/or effects of the seed aerosol. To clarify this, we have rephrased this original sentence to:

"*Correspondingly, to account for any potential chamber background and/or seed effects, the particle phase data collected during the "dark unreactive" phase (i.e., background filter collected when the chamber was filled with the VOC(s) and seed particles in the dark) were subtracted from the particle phase measurements.*"

12. line 236-238: "The resultant volatility of the identified compounds in each system was expressed in logarithmically spaced bins in the VBS framework as:…" Does this mean that the direct volatility information of SOA total bulk was not used or analysed? This could give a direct way to compare different experiments with each other, assuming proper instrument calibration.

In this work, we only show results from the compounds we have identified in the FIGAERO-CIMS mass spectra; this corresponded to ≥70% of the total signal (see L200). The remaining, unidentified fraction was largely comprised by compounds with low signal-to-noise ratio (for further, detailed information the reader is referred to our comment response #24 in RC1 comments in Voliotis et al., 2021). Therefore, we believe that the reported volatility distributions presented here are representative for the bulk of the SOA that can be detected by the FIGAERO-CIMS.

We are unsure of what the reviewer is suggesting further here.

13. lines 328-329: "Although the SOA mass formed at the isoprene single-precursor experiments was found to be below our background (~1 μg m$^{-3}$), in all isoprene-containing systems studied here we were able to attribute a small fraction of the total FIGAERO-CIMS signal (≤6%) to isoprene-derived products" Here it is mentioned that isoprene-derived products were found in mixed systems. Were the isoprene-derived products identified only form these mixed experiments, or was the isoprene only experiment data analysed to identify the isoprene-derived

products? How easy were they to distinguish from whatever was in the "background" particles produced in the chamber?

As with all the mixtures, we used the products identified in the isoprene only experiments to identify the isoprene-derived products in the mixtures. In every experiment, we identified the dominant compounds in the combined averaged gas and particle phase mass spectrum. In the isoprene-only experiments, those compounds were predominantly found in the gas phase whereas in the mixtures they appeared to have stronger contributions in the particle phase, likely due to the increase in the absorptive mass (see L331-332). Adopting the peak fitting and background subtraction methods described in the responses above (see responses to comments #9 and 11), all the isoprene-derived peaks that are reported here had high signal-to-noise ratio (S/N>2) and were above the chamber/seed background signals.

14. lines 332-335: "In support of this, it can be seen from Figures 4 and 5 that the majority of the isoprene-derived products have $C^* \geq 100$ µg m-3 which would cause these products to remain predominantly in the gas phase in the single-isoprene experiments where the total absorptive mass was $\leq 1$ µg m-3." In Table 1 it can be seen that in high [isoprene] and [α-pin]+[isoprene ] the seed is comparable. Is the idea here that the isoprene oxidation products would only condense towards the end of the mixed experiment? Is this supported by the data during the experiment? I.e. theis more isoprene-derived products that only appear at the end of the experiment(s)?

Indeed, the seed concentration at the beginning of each experiment was comparable. The inorganic seed concentration was at its highest at the beginning of each experiment and decayed over time in a similar fashion in all the experiments conducted. The SOA produced from the oxidation of the precursors would then partition to the condensed phase according to their vapour pressure and the amount of the available absorptive mass (e.g., Pankow, 1994). In the isoprene single experiment the SOA condensed on the seed particles were negligible (below our background) whereas in the α-pinene+isoprene SOA condensed rapidly after the initiation of the experiment reaching tens of µg m$^{-3}$ (see Fig. S3). An interesting question here is what is considered as absorptive mass. The effectiveness of an inorganic seed particle opposed to a heavily coated with organics (i.e., single isoprene vs. α-pinene+isoprene) to act as absorptive mass is debatable (see L231-236). Traditionally, the absorptive partitioning theory considers that only the organic fraction of the particles can be accounted as an effective medium for the partitioning of the organics in the particle phase (e.g., Pankow, 1994; Donahue et al., 2006). More recent studies have showed that inorganic seeds can potentially act as an effective medium (Zhang et al., 2014), however to an unknown extent compared to the organics. Here, our interpretation is that the significantly organic-containing particles that are formed in the mixed systems, along with the higher total mass

concentrations at the end of the experiments compared to the beginning, are favouring the partitioning of the more volatile isoprene-derived products in the condensed phase (L331-332).

15. line 342: "“Common” were classified as the products with common elemental formulae between the systems involved." Does this mean all three cases of systems? And not just the ones used in each experiment?

We thank the reviewer for this suggestion. Indeed, the "common" were referring to the common compounds between the products derived from the precursors used in each system. The sentence has now been rephrased to: *"Common" were classified as the products that were identified in each mixture as well as at the respective single precursor experiments of that mixture.*

16. line 361: relating Figure 4: there doesn't seem to be $C_{15}$ compounds in α-pinene+isoprene mix, although one might expect $C_{10}$ and $C_5$ compounds to form $C_{15}$ Is there a possible explanation for this?

Indeed, one might expect a number of $C_{15}$ compounds in the a-pinene/isoprene mixed system from the cross reactions of $C_{10}$ and $C_5$ radicals. There are a few reasons why they might be absent from our analysis.

First, their absence might be related to our peak identification protocol. In each system, we conducted the fitting process in the 190-550 m/z range and sorted the UMR mass spectrum in descending order, based on the signal intensity. We then started the peak identification from the UMR peaks with the highest intensity towards those with the least. After we fitted ≥70% of the total signal (excluding the signal of the reagent ions), the peak identification was challenging and in many cases without reasonable compounds within our accepted fitting error range (~≤6 ppm; L198-200). Therefore, a possible explanation for this might be that these compounds were contributing too little to the total signal so we did not identify them. Alternatively, as also the reviewer points out in various comments, the larger molecules might be more liable to thermal decomposition so a considerable fraction of them might be lost. These might also explain the generally low signal fraction of the $C_{>10}$ compounds identified here (4-5%).The above discussion has been added in the revised manuscript.

17. line 377-378: "The SOA volatility distribution measured in the o-cresol/isoprene showed high variability between the two experiments conducted (i.e., see error bars in Fig. 3 middle panels)…" For me all the error bars in each panel seem comparable in range (top and bottom, respectively). I see no real difference in the middle panels compared to the other two.

We agree with reviewer that this might have been over-stated, particularly for the top panels. The fraction of the products that were unique to this mixture however (bottom panel), showed considerably higher variability compared to all the other systems (particularly in the C*=100 and 1000 µg m$^{-3}$ bins). We have altered the discussion in the revised manuscript to account for this.

18. line 371: "…TD showed three peaks,…" I'm not sure how you can say there are three peaks in the TD figures. With such large error bars, it could be as justifiable to include the signal at 0.1 as is to include the 100 µg m$^{-3}$.

True. This sentence has been omitted from the revised manuscript.

19. lines 374-376: "The SOA concentration in these experiments was of the order of 10 µg m-3 so the presence of such high concentrations of IVOCs in the particulate phase can be explained either by very high corresponding IVOC concentrations in the gas phase or by a significant underestimation of the less volatile SOA components by the FIGAERO-CIMS technique." Has it been considered if the high IVOC contribution might be an effect from using FIGAERO-CIMS, and assigning false C* to the smaller thermal decomposition compounds known to appear id FIGAERO-CIMS data?

We thank the reviewer for this suggestion. Indeed, thermal decomposition might be a significant factor that could be affecting this system to a higher extent than the others, leading to a false assignment of compounds with higher C* and thereby leading to particularly higher IVOC fractions. Yang et al. (2021) provided a parametrisation to estimate the fraction of the signal that could potentially be attributed to considerably thermally decomposed products based on the double bond equivalent (DBE>2), the nO (nO>4) and $T_{max}$ ($T_{max}$> 72°C) for a wide range of compounds with different functionalities. Adopting this parametrisation and applying it to our data and restricting it to the IVOC fraction, we find that in all *o*-cresol containing mixed systems ~0.1-3.0% of the total IVOC signal could be attributed to considerably thermally decomposed products (0.5-3.0% for the o-cresol/isoprene; 1.6-2.5% for the o-cresol/α-pinene and 1.1-1.9% for the ternary). Therefore, this suggests that the thermal decomposition that might be occurring in the *o*-cresol/isoprene system is not significantly higher than any other *o*-cresol-containing system and cannot explain the particularly high IVOC fraction that was observed. Of course, it should be noted that the parametrisation provided by Yang et al. (2021) has been derived with a particular instrument under certain operation conditions and might not be universally applicable. However, it might still be used as a relative comparison between the systems. We have added this discussion in the revised version of the manuscript.

20. line 379: what does "appreciable" mean here? How much of the 46% was this largest fraction?

The wording of "appreciable" was mainly derived based on the visual inspection of the figures, so we thank the reviewer for his/her suggestion to be more quantitative. The fraction of the unique of the mixture products that had nC>8, O:C≥0.7 and moderate OSc~0.6 was <5%. In the revised manuscript we have altered slightly the above criteria to reflect a higher fraction of the signal (~19%) to nC≥8 and O:C≥0.7.

21. lines 416-418: "Considering that in the binary systems, only half of the total initial reactivity of the system was attributed to each of the precursors, the experiments at half initial reactivity were used to make such predictions." Here again bringing up if OH exposures were calculated/estimated for each system? This could help with understanding how the system developed, and present a possible additional source for observed differences.

Please see our responses to the comments #2 and 8. The estimated OH was comparable between all systems. However, in our systems we had combined OH and $O_3$ oxidation. This could lead to changes in the chemical regime between the experiments and make challenging the systems' comparison (see Voliotis et al., 2022).

22. line 422: "…in the FIGAERO-CIMS measurements the LVOC, SVOC and IVOC content were about 0, 87 and 13%, respectively (Fig. S4)…" Relating also for the mentioned Fig. 4S: how much of the IVOC signal, present only in FIGAERO-CIMS data, is actually thermal decomposition? as this been considered and taken into account during data analysis? The total lack of LVOC in FIGAERO-CIMS data seems odd, as their presence is shown in TD data.

We agree that the thermal decomposition of the products in the FIGAERO can be undoubtedly important and can potentially contributing to the IVOC signal. We also acknowledge that the effect of the potential effect of the thermal decomposition has not been discussed enough in the original manuscript and in the revised manuscript we refer to it in various places as potential explanations. Adopting the same logic as the one described in our response to the comment #19 to the remaining systems we find that the potential contribution of the considerably thermally decomposed products was 0-3%. Again, this parametrisation could only be indicative here as it might only be related to the specific instrument used in the Yang et al. (2021).

The absence of LVOC seems indeed surprising, given that compounds with such low C* have been previously measured with the FIGAERO-CIMS (e.g., Tikkanen et al., 2020). However, in our case, this was not the case. As we show in Voliotis et al., (2021), regardless the volatility calculation method ($T_{max}$ vs. $V_p$ calibrations or partitioning calculations), we were not able to reliably identify substantial fractions of compounds in the LVOC fraction. However, as we further mention in our response to #3 and discuss in detail in Voliotis et al., (2021), we did show that our approach can be used to provide

representative changes of the volatility between the different systems, despite the challenges in the quantification (see L286-291).

23. lines 441-445: "Particularly, the FIGAERO-CIMS measurements showed that the o-cresol/isoprene system had a ~10% increased measured than predicted IVOC fraction with a corresponding decrease in the SVOC fraction. The TD measurements showed a similar LVOC/SVOC split between the measured and predicted volatilites, yet the measured volatility distributions exhibited significantly higher contributions in the $C^* \geq 100$ µg m-3bin (18% vs. 11% of the total mass; Fig. 3)." Why the discrepancy between FIGAERO-CIMS and TD?" Cresol-isoprene TD has larger variability compared to the other two systems, but much less than in FIGAERO-CIMS. Could this be related to formed products breaking more easily in the cresol-isoprene case than e.g. α-pinene-isoprene case? This would explain the large number of IVOCs present, as it has been suggested that compounds labelled as IVOC can sometimes be actually thermal decomposition products. This might go towards explaining some of the changes in predicted vs measured IVOC/SVOC..

Again, we refer the reviewer to our responses in the comments #19 and 22. Briefly, the discrepancy between the volatility distributions between the FIGAERO-CIMS and TD can be attributed to multiple reasons as for example, that the TD results are expressed as a function of the total SOA particle mass whereas the FIGAERO-CIMS as a fraction of the total signal of the compounds that could be identified, assuming uniform sensitivity (L184-186). Further, the signal-to-noise limitations in the calculations of the volatility using the partitioning approach can limit the low and high $C^*$ values. In our previous, companion works (see Du et al., 2021;Voliotis et al., 2021) as well as in other studies conducted previously (e.g., Stark et al., 2017) these are discussed in further detail and we are referring to them in the original manuscript (see L286-291). Further, according to our estimations in the responses #19 and 22, the potential fraction of the considerably thermally decomposed products in the o-cresol/isoprene system is not significantly different than those observed in any other single or mixed system. Therefore, despite thermal decomposition could be affecting all of our systems to a truly, unknown extent, we do not have any indications to suggest that a particular system might be different than others and explain the considerable differences in the observed volatility. As we mentioned above, indeed, the potential impact of thermal decomposition might not have been discussed enough in the original manuscript and we have added some discussion on it in the revised manuscript.

24. lines 489-490: "…were almost identical with those measured (≤3% of the total mass/signal; Fig. S5 and S6)." Figures S6b and Fig 3 tern.mix do not seem identical. Am I comparing the right ones? This is somewhat unclear.

In this sentence we compare the measured volatility in the α-pinene/o-cresol system (Fig. S5) with the one projected at the same absorptive mass of the ternary mixture (Fig. S6). As we state in the sentence in L489, the difference in the volatility distribution between Fig.S5 and S6 is negligible. Indeed, as we also state in the L485 the volatility of the ternary mix (Fig. 3) and the α-pinene/o-cresol system (Fig. S6) is considerably different. In the revised manuscript, we have modified these sentences to be clearer.

25. lines 493-494: "…a system that cannot also be predicted based on the additivity (see Fig. S5)." Should this say "a system that can also"? They seem relatively consistent with predicted to me.

The differences in the measured vs. the predicted volatilities in this system are in the same order as those observed in the o-cresol/isoprene and ternary mixtures (i.e., ~10% of the total signal and ~5% of the total mass; see caption of Fig. S5). Therefore, we believe that it warrants to be classified as a system where the volatility could not be predicted, opposed, for example, the α-pinene/isoprene system, where the differences between measured and predicted volatility were negligible. We see that the discussion might have been somewhat confusing in this part therefore we have attempted to rephrase our statements in the revised manuscript.

26. lines 500-503: "Interestingly, the measured volatility distributions are lower compared to those predicted in all cases and for both techniques, further confirming that the chemical interactions occurring in this system will lower the resultant SOA particle volatility." The measured and predicted values presented in S7 middle panels seem very similar in both methods. Is it really worse than what is shown in Figure 3 for ternary system? I cannot see a real difference.

Indeed, the measured volatility distribution in the ternary system is not significantly different than those predicted based on the alternative approach shown in the middle panels of Fig. S7. Still, the measured volatility distributions however, show an ever so slightly increased fraction of lower volatility compounds compared to the predictions. This is now explicitly mentioned in the revised manuscript.

Our point here is that there are multiple ways to consider the predictions of the volatility in mixed precursor systems and likely, none of them is correct. The results from our overall programme of work (e.g., see Voliotis et al., 2022) along with the results presented here, demonstrate that the SOA formation potential and properties can change upon mixing various VOC precursors and those changes cannot be predicted assuming additivity. This highlights the need for further mechanistic understanding of the behaviour of such systems.

27. line 556: "Such an example might be clean environments where the RO2:HO2 has been found to be high…" Would such clean atmosphere have the quite high [VOC] used in these experiments?

Indeed, the concentrations used in our study were not atmospherically relevant. The example given was only indicative to refer to environments where the chemical regime of the real atmosphere could be $RO_2$-$RO_2$ dominated, such as in our study as a result of the high VOC concentrations used. We have added additional information in this section according to our response to the comment #1.

Technical comments:

We thank the reviewer for his/her technical suggestions. They have all been addressed in the revised manuscript. The comments below that required additional information/clarification have been addressed individually.

28. line 17: "using" instead of "use of a"

29. line 48: "in a chemically highly complex" instead of "in a highly chemically complex"

30. line 177: "at the exhaust of the ion molecule region (IMR)" what does this mean? I do not quite understand where the mass flow controller was positioned.

A mass flow meter is positioned downstream of the IMR pump (i.e., at its exhaust) to measure in real time the flow passing through the IMR region. This was to ensure that there were no blocks as well as to ensure that the total flow passing was comparable between the gas and particle phases. This sentence has been modified to be clearer.

lines 179-180: "The reagent ions were produced by passing CH3I and UHP $N_2$ over a 210 Po radioactive source introduced directly into the IMR." would be better to replace "introduced" with "connected".

31. lines 329-330: "…is significant and/or this is could be attributed to a" please fix the grammar

32. lines 332: "…opposed to the single-precursor isoprene would have favoured the partitioning of the more volatile species…" grammar: add "which" between "isoprene" and "would"

33. line 338: " Figure 3: Measured and predicted based on the additivity SOA particle volatility distributions from the TD…" the "what" of the description should be at the start, after "predicted".

34. line 338: relating Figure 3: it is commendably to have the error bars in the figures, but they make reading the bottom row bar plots quite tricky. There are already many coloured segments in the plots, with the added error bars especially bottom left figure is almost unreadable in detail. One option could be to ,for this one figure, to leave the error bars out form the main figure, but then include them in additional Supplements figure.

We thank the reviewer for this suggestion. In the revised manuscript we have removed the error bars from the bottom panels of this plot and have added another in the SI containing the errorbars.

35. line 348: add "were" after "6%"

36. line 350: "and our technique is unable to resolve" should this be "OR our technique is unable to resolve"? Add "that" before "our technique".

37. lines 392: relating Figure 5: the chosen colour scheme might be very challenging to readers with red-green colour blindness.

We thank the reviewer for raising this. In this work and all of our companion papers (see Voliotis et al., 2021, 2022; Du et al., 2022), we have selected our three single precursor systems to be presented as the three primary colours (blue, red, yellow for α-pinene, o-cresol and isoprene, respectively), each binary mixture as the secondary colours that derive from each combination of the respective primary colours and so on (e.g., green for α-pinene/isoprene). Particularly for the ternary system (i.e., Fig. 5), the categories that are resulting from our formulae separation technique are higher than the maximum amount of colours that recommended for a colour-blind friendly palette (11 vs. 8 colours). This makes quite challenging the selection of alternative colour or symbol patterns. In order to keep the consistency between all of our companion papers, we will not be changing the colour scheme of Figure 5.

38. line 516: "to what extent" rather than "the extent that"

39. line 517: "combination of both" instead of "combination of the above".

40. line 526: "interactions still lead" not leads

41. Supplementary material:

42. Figure S2: Are the symbols presenting particle phase signal, or the square root of the signal, as in all other figures?

They representing the square root of the signal, this is clarified in the revised SI.

43. Figure S3: Is isoprene shown in the figure? I can't see it.

Isoprene is shown but its signal was zero. We agree that there is not point showing it, so it has been removed from the revised version of the figure.

44. Figure S5: last sentence of the description should read: "…result that is consistent **with** the chemical information…"

**References**

Cao, Z., Zhou, X., Ma, Y., Wang, L., Wu, R., Chen, B., and Wang, W.: The Concentrations, Formations, Relationships and Modeling of Sulfate, Nitrate and Ammonium (SNA) Aerosols over China, Aerosol and Air Quality Research, 17, 84-97, 10.4209/aaqr.2016.01.0020, 2017.

Donahue, N. M., Robinson, A. L., Stanier, C. O., and Pandis, S. N.: Coupled Partitioning, Dilution, and Chemical Aging of Semivolatile Organics, Environmental Science & Technology, 40, 2635-2643, 10.1021/es052297c, 2006.

Du, M., Voliotis, A., Shao, Y., Wang, Y., Bannan, T. J., Pereira, K. L., Hamilton, J. F., Percival, C. J., Alfarra, M. R., and McFiggans, G.: Combined application of Online FIGAERO-CIMS and Offline LC-Orbitrap MS to Characterize the Chemical Composition of SOA in Smog Chamber Studies, Atmos. Meas. Tech. Discuss., 2021, 1-42, 10.5194/amt-2021-420, 2021.

Lee, B. H., Lopez-Hilfiker, F. D., Mohr, C., Kurtén, T., Worsnop, D. R., and Thornton, J. A.: An Iodide-Adduct High-Resolution Time-of-Flight Chemical-Ionization Mass Spectrometer: Application to Atmospheric Inorganic and Organic Compounds, Environmental Science & Technology, 48, 6309-6317, 10.1021/es500362a, 2014.

Pankow, J. F.: An absorption model of the gas/aerosol partitioning involved in the formation of secondary organic aerosol, Atmospheric Environment, 28, 189-193, https://doi.org/10.1016/1352-2310(94)90094-9, 1994.

Schobesberger, S., D'Ambro, E. L., Lopez-Hilfiker, F. D., Mohr, C., and Thornton, J. A.: A model framework to retrieve thermodynamic and kinetic properties of organic aerosol from composition-resolved thermal desorption measurements, Atmos. Chem. Phys., 18, 14757-14785, 10.5194/acp-18-14757-2018, 2018.

Shao, Y., Wang, Y., Du, M., Voliotis, A., Alfarra, M. R., O'Meara, S. P., Turner, S. F., and McFiggans, G.: Characterisation of the Manchester Aerosol Chamber facility, Atmos. Meas. Tech., 15, 539-559, 10.5194/amt-15-539-2022, 2022.

Stark, H., Yatavelli, R. L. N., Thompson, S. L., Kang, H., Krechmer, J. E., Kimmel, J. R., Palm, B. B., Hu, W., Hayes, P. L., Day, D. A., Campuzano-Jost, P., Canagaratna, M. R., Jayne, J. T., Worsnop, D. R., and Jimenez, J. L.: Impact of Thermal Decomposition on Thermal Desorption Instruments: Advantage of Thermogram Analysis for Quantifying Volatility Distributions of Organic Species, Environmental Science & Technology, 51, 8491-8500, 10.1021/acs.est.7b00160, 2017.

Tikkanen, O. P., Buchholz, A., Ylisirniö, A., Schobesberger, S., Virtanen, A., and Yli-Juuti, T.: Comparing secondary organic aerosol (SOA) volatility distributions derived from isothermal SOA particle evaporation data and FIGAERO–CIMS measurements, Atmos. Chem. Phys., 20, 10441-10458, 10.5194/acp-20-10441-2020, 2020.

Voliotis, A., Wang, Y., Shao, Y., Du, M., Bannan, T. J., Percival, C. J., Pandis, S. N., Alfarra, M. R., and McFiggans, G.: Exploring the composition and volatility of secondary organic aerosols in mixed anthropogenic and biogenic precursor systems, Atmos. Chem. Phys., 21, 14251-14273, 10.5194/acp-21-14251-2021, 2021.

Voliotis, A., Du, M., Wang, Y., Shao, Y., Alfarra, M. R., Bannan, T. J., Hu, D., Pereira, K. L., Hamilton, J. F., Hallquist, M., Mentel, T. F., and McFiggans, G.: Chamber investigation of the formation and transformation of secondary organic aerosol in mixtures of biogenic and anthropogenic volatile organic compounds, Atmos. Chem. Phys. Discuss., 2022, 1-49, 10.5194/acp-2021-1080, 2022.

Wu, J., Brun, N., González-Sánchez, J. M., R'Mili, B., Temime Roussel, B., Ravier, S., Clément, J. L., and Monod, A.: Substantial organic impurities at the surface of synthetic ammonium sulfate particles, Atmos. Meas. Tech., 15, 3859-3874, 10.5194/amt-15-3859-2022, 2022.

Yang, L. H., Takeuchi, M., Chen, Y., and Ng, N. L.: Characterization of thermal decomposition of oxygenated organic compounds in FIGAERO-CIMS, Aerosol Science and Technology, 55, 1321-1342, 10.1080/02786826.2021.1945529, 2021.

Ylisirniö, A., Barreira, L. M. F., Pullinen, I., Buchholz, A., Jayne, J., Krechmer, J. E., Worsnop, D. R., Virtanen, A., and Schobesberger, S.: On the calibration of FIGAERO-ToF-CIMS: importance and impact of calibrant delivery for the particle-phase calibration, Atmos. Meas. Tech., 14, 355-367, 10.5194/amt-14-355-2021, 2021.

Zhang, X., Cappa, C. D., Jathar, S. H., McVay, R. C., Ensberg, J. J., Kleeman, M. J., and Seinfeld, J. H.: Influence of vapor wall loss in laboratory chambers on yields of secondary organic aerosol, 111, 5802-5807, 10.1073/pnas.1404727111 %J Proceedings of the National Academy of Sciences, 2014.

---

## Author Comment (AC2)

SOA is important to both air quality and climate. Most previous studies predicted the SOA mass based on the SOA yield method from a signal precursor. However, VOCs and their oxidation products in the real atmosphere are a complex mixture (Nie et al., 2022). The interactions among different VOC oxidation processes can potentially influence the SOA yield but are largely understudied. This study investigated the influence of isoprene on the SOA formation from a-pinene, o-cresol, and their mixtures. The results did provide useful information and fit the scope of ACP. However, I have a few comments before this manuscript can be published.

We'd like to thank Referee 1 for their positive comments and to respond to the general and detailed comments as follows (reviewer comments in black and our responses in blue; the line numbers referred throughout are referring to the original manuscript).

1. **Reliability of measurement technology:** Thermal denuder and FIGAERO-CIMS need to heat the sample before detection. This may induce interferences via thermal decomposition, as well as chemical reactions occurred during the heating. For example, in the α-pinene/isoprene mixture system, there were no detected product molecules with nC=15 (Fig. 4a), which should be one crucial group of products from the cross-reactions between C10-RO2 and C5-RO2. In addition, most detected molecules by FIGAERO-CIMS were less oxidized (lower O:C) than those observed in the gas phase. What's the possible reason? And how to evaluate these possible instrument-induced interferences?

We thank the reviewer for his/her suggestion to discuss in more detail the potential effect of thermal decomposition in our manuscript. This has been also raised from the reviewer #2 and we acknowledge that it warrants some additional discussion. Indeed, thermal decomposition is undoubtedly affecting our measurements to an unknown extent (Stark et al., 2017). To reflect on this, we have added some discussion in the revised manuscript to acknowledge the thermal decomposition as a potential measurement artefact.

Regarding $C_{15}$ compounds, indeed, one would expect to be potentially important group of compounds and their absence may be related to the thermal decomposition. Alternatively, these products might had very low signal, so we couldn't identify them reliably. This has been explicitly mentioned in the revised manuscript.

In response to "*most detected molecules by FIGAERO-CIMS were less oxidized (lower O:C) than those observed in the gas phase*"; in this work, we did not show any comparative O:C results between the gas and particle phase measurements. However, in our previous companion manuscript (see Voliotis et al., 2021), we contrasted gas and particle phase measurements from a subset of the experiments presented here and we found higher sum signal fractions of compounds with high O:C (O:C>0.7) in the particle

than in the gas phase in the o-cresol system (24 and 9%, respectively). However, the opposite was also true for the more volatile α-pinene system (12 and 25%, respectively), while their mixture was in-between (i.e., α-pinene/o-cresol; 23 and 24% respectively), as was its particle volatility. Therefore, based on our observations, we believe that the split of O:C between the gas and particle phase is largely attributable to the volatility of the systems.

We see however the point that the reviewer is trying to make, i.e., that thermal decomposition of the products in the particle phase mode might result in lower observed O:C in the particle phase than in the gas phase, however in our case, volatility seemed to be a stronger driver for this. Again, as it was mentioned above, the potential effect of thermal decomposition to our results has been added as potential explanation in the discussion of the revised manuscript.

2. **Calculation method of partitioning coefficient (Line 230)**: This method is only valid when a compound is in equilibrium between the gas and particle phases and cannot be used to assess the volatility of very condensable compounds (< LVOC). Moreover, considering the large number of potential gas- and particle-phase reactions, it is difficult to ensure that compounds with the same molecular formula in both phases are indeed the same molecule, i.e., have the same molecular structure. The authors need to provide more results to clarify the accuracy, uncertainty and applicability of the method. Given the use of FIGAERO-CIMS, the authors may be able to infer volatility based on thermograms for individual molecular components (Thornton et al., 2020), and may be able to obtain information by comparing the results of the thermal-desorption-based approach with the results of the equilibrium method used in the text.

A similar point have been raised from the reviewer #2 (see comment #3), so we acknowledge that more information is required about this. In the revised manuscript we have added more information about our reasoning for selecting this method, along with its potential limitations.

Indeed, the partitioning calculations have an inherent assumption of gas/particle equilibrium. As we further show in Voliotis et al., (2021), Du et al. (2021) and also previously shown by Stark et al. (2017), this approach has some limitations in quantifying compounds with very low or high volatility (see L286-291). Previously, we have contrasted the partitioning approach with the thermogram-based using explicit $T_{max}$-$V_p$ calibrations as well as the TD-AMS measurements for a subset of the systems (see Voliotis et al., 2021). We found that the volatility derived from partitioning calculations were more realistic and broadly comparable with the concurrent TD-AMS measurements opposed to the thermal desorption/calibration based approach. Hence that exercise informed our decision to select the partitioning approach here.

In more detail, as we state in our response to the reviewer's 2 comment #3: "*The volatility distributions obtained from the $T_{max}$ calibrations were unrealistically high with the vast majority of the signal being accumulated in $C^*>100$ µg m$^{-3}$ at SOA particle mass loadings $>200$ µg m$^{-3}$, inconsistent with the absorptive partitioning theory. We attributed this to the method we selected to introduce the calibrants in the instrument (syringe method) that was more recently shown to have an effect on the $V_p$-$T_{max}$ relation (Ylisirniö et al., 2021), as well as to unquantified matrix effects during the desorption (Schobesberger et al., 2018). On the other hand, the volatility distributions obtained from the partitioning calculations were realistic and broadly comparable with the concurrent TD-AMS measurements. Importantly, despite the challenges in the volatility quantification from the FIGAERO-CIMS, we showed that our method is able to capture the volatility changes between the systems. Therefore, in this companion and follow-up paper, we have selected the partitioning method to illustrate the volatility changes and their effect in the volatility predictions as a result of the mixing of the precursors. The application of this method to our data is detailed in section 2.5 of the original manuscript.*"

A brief version of the above has been added in the revised manuscript in the methods section.

3. FIGAERO-CIMS can also measure the gas-phase oxidation products. Why not add some discussions of gas-phase chemistry to support the interpretation of the aerosol phase observation?

Indeed, FIGAERO-CIMS can provide detailed information both for the gas and particle phase products. Owing to our selected method to calculate the particle volatility that requires both gas and particle phase measurements to calculate the partitioning coefficient of each product(s) (see L227-234), we have used one common peaklist for both the gas and particle phase measurements and the only thing varying between them is their contributions in either phase. Therefore, the oxidation products that are shown here (Fig. 1, 2, 4 and 5) are broadly reflecting both the gas and particle phase composition, and each systems' volatility is by extent providing some information about their ratio. Consequently, the interpretation of our systems based on the volatility from the partitioning approach as well as an illustration of the particle composition should be adequate for the purposes of this work, which is to explore the volatility changes in mixed systems from the addition of isoprene and to assess whether additivity can predict those changes.

Nonetheless, we agree with the reviewer that the results that can be provided by the gas and particle phase measurements from the FIGAERO-CIMS can be insightful in understanding the systems' behaviour. In a subsequent manuscript (Du et al., in prep), we apply the methodology developed in Du et al. (2021) to all systems and by utilising dimensionality reduction techniques, we are able to identify the drivers of the SOA formation from the gas and particle phase time-series and link them with the results presented here.

4. **To Line 298-300**: The carbon number is not shown in Fig. 1c and Fig. S1.

We thank the reviewer for pointing this out. This has been corrected in the revised manuscript.

5. Table 1: Exp. 1-4: why the SOA yield change from 0.32 to 0.15?

The SOA particle yield is reducing as the initial VOC concentration reduces (see column 4 of table 1); experiments at full initial reactivity (i.e., 309 ppb; Exp. 1-2) have higher SOA particle yield (Y=0.32) than those at 1/3 initial reactivity (i.e., 103 ppb; Y=0.15; Exp. 4). According to previous works, this is an expected behaviour, e.g., see Alfarra et al., (2012) and Chen et al., (2019), and references therein.

6. **To Line 357, 380, 408-409**: Based on the assigned elemental formulas, many compounds seem to be not closed-shell molecules, e.g. $C_{10}H_{10}NO_4$ and $C_{12}H_{11}O_4$, etc. Can the authors supply the results of HR peak fitting (including the measured signal, the fitted peak and the residual)?

We thank the reviewer for pointing this out. Indeed, the observation of open-shell species is unlikely in the FIGAERO-CIMS. We have now conducted a thorough investigation in our peak assignment and in retrospect, we are confident for our assignments, as they were the most likely candidates from all the related suggestions within our accepted error range. For example, see the case of $C_{12}H_{11}O_4$ below, a compound detected in the *o*-cresol/isoprene system including measured signal, fitted peaks and residuals. What we didn't account for, was any potential secondary chemistry in the IMR (e.g., Zhang and Zhang, 2021) and/or potential adducts with $I.H_2O$, $I.O_3$, etc (e.g., Veres et al., 2020;Murschell et al., 2017). As the compounds shown in the original manuscript were randomly chosen to cover a range of nC and did not reflect their importance, in the revised manuscript we selected to show the unique compounds that had the highest signal contributions.

**Case of $C_{12}H_{11}O_4$**

The figure below shows the fitted peaks at the UMR of the related ion (i.e., $C_{12}H_{11}IO_4$; 346 m/z) directly exported from Igor Pro, using the Tofware v. 3.1.2 workflow. Three peaks were fitted in the UMR, shown in different colours and the total fit is shown in black. Here, we attempt to identify the largest peak (shown in green; exact m/Q: 345.96976). The table below shows all the suggested molecular formulae and their exact mass, sorted based on the fitting error (ppm). As can be seen, within our trusted error range (~6 ppm), the only two suggested formulae that can exist as molecules are the $C_7H_{11}IN_2O_6$ (5.2 ppm) and the $C_{12}H_{11}IO_4$ (6.4 ppm); both open-shell species. This remains to be the case even if we extend our trusted fitting error range to ~9 ppm. Based on these elemental formulae, the former, unsaturated molecule containing two nitrogen atoms, is probably unlikely in this system (*o*-cresol/isoprene). Therefore, our only viable option in this case would be to assign the $C_{12}H_{11}IO_4$.

[Figure]

| Suggested Formula | Exact m/Q | Error (ppm) |
|---|---|---|
| $C_{10}H_4NO_{13}$ | 345.9682641881199 | -0.83299005 |
| $C_2H_8N_3O_{15}S$ | 345.9676127394399 | 1.0499843 |
| $C_{17}H_2N_2O_5S$ | 345.96844234566 | -1.3479427 |
| $C_3H_{24}I_2S$ | 345.96881574792 | -2.4272358 |
| $C_{10}H_9IN_3O_3$ | 345.96886306207 | -2.5639939 |
| $C_4H_{10}O_{16}S$ | 345.9689554103001 | -2.8309195 |
| $CH_{17}INO_9S$ | 345.96687420311 | 3.184689 |
| $C_4H_{15}IN_2O_6S$ | 345.96955428425 | -4.5619168 |
| $C_7H_{11}IN_2O_6$ | 345.9661829809299 | 5.1826425 |
| $C_{14}H_4NO_8S$ | 345.96576226452 | 6.3987126 |
| $C_{12}H_{11}INO_4$ | 345.97020573293 | -6.44487 |
| $C_7H_6O_{16}$ | 345.96558410698 | 6.9136734 |
| $C_6H_{20}I_2$ | 345.9654444446 | 7.3173647 |
| $C_{13}H_2N_2O_{10}$ | 345.97094426926 | -8.5795336 |
| $C_{20}N_3O_2S$ | 345.9711224268 | -9.0944777 |

7. **To Line 449-450**: It seems that SOA particle mass yield of o-cresol was reduced by the presence of isoprene (from 0.11 to 0.06/0.05).

We thank the reviewer for pointing this out, indeed the yield was reduced in this mixed system compared to the single precursor experiment. We have corrected our statement in the revised manuscript.

8. **To Line 541**: As shown in the study of McFiggans et al. (2019) and Heinritzi et al. (2020), isoprene depletes OH radicals, preventing their reaction with monoterpenes, and the resulting isoprene peroxy radicals scavenge highly oxygenated monoterpene products. These effects ultimately suppress both particle number and mass of secondary organic aerosol. It can be simply inferred that the mass yield, composition, and volatility of SOA will be altered after mixing isoprene in the monoterpene system, but this was not the case in this study. Given that FIGAERO-CIMS can provide near-molecular information, it is appropriate for the authors to discuss the above situation.

Indeed, isoprene can compete for the available oxidants and/or scavenge the peroxy radicals of the other precursors involved in each system (e.g., $o$-cresol for the system described in L541). This can result a reduction of the SOA mass, which was observed here (see response to our comment #7 above and Table 1). Further, in all mixed systems we found evident changes in the chemical composition compared to the single precursor systems, which is discussed in the original manuscript submission in detail (e.g., see L430-439, 454-464, L510-519). We similarly found changes in the SOA particle volatility (see Sections 4.1 and 4.2), apart from the α-pinene/isoprene system, consistent with previous literature (Ylisirniö et al., 2020; see L428).

More specifically, in all mixed systems, we attributed a significant fraction of the signal in products that were uniquely found in mixtures (see L32). These products might either derive as $RO_2$-$R'O_2$ termination products (where R and R' radicals deriving from each individual precursor in each mixed system) and/or as products that were derived from the alteration in the oxidation pathways of each precursor due to the potential changes in the chemical regime (see Voliotis et al., 2022). We cannot distinguish which of the two is more important as FIGAERO-CIMS cannot provide molecular information, however the likelihood of both to affect our results is acknowledged in the original manuscript e.g., L451.

Given the above, we are unsure of what the reviewer is suggesting here.

**Reference**

Nie, W., et al., Secondary organic aerosol formed by condensing anthropogenic vapours over China's megacities. Nature Geoscience, 10.1038/s41561-022-00922-5, 2022.

Heinritzi, M., et al., Molecular understanding of the suppression of new-particle formation by isoprene, Atmos. Chem. Phys., 20, 11809-11821, 10.5194/acp-20-11809-2020, 2020.

McFiggans, G., et al., Secondary organic aerosol reduced by mixture of atmospheric vapours, Nature, 565, 587-593, 10.1038/s41586-018-0871-y, 2019.

Thornton, J. A., et al., Evaluating Organic Aerosol Sources and Evolution with a Combined Molecular Composition and Volatility Framework Using the Filter Inlet for Gases and Aerosols (FIGAERO), Accounts of Chemical Research, 53, 1415-1426, 10.1021/acs.accounts.0c00259, 2020.

**References**

Alfarra, M. R., Hamilton, J. F., Wyche, K. P., Good, N., Ward, M. W., Carr, T., Barley, M. H., Monks, P. S., Jenkin, M. E., Lewis, A. C., and McFiggans, G. B.: The effect of photochemical ageing and initial precursor concentration on the composition and hygroscopic properties of β-caryophyllene secondary organic aerosol, Atmos. Chem. Phys., 12, 6417-6436, 10.5194/acp-12-6417-2012, 2012.

Chen, T., Liu, Y., Chu, B., Liu, C., Liu, J., Ge, Y., Ma, Q., Ma, J., and He, H.: Differences of the oxidation process and secondary organic aerosol formation at low and high precursor concentrations, Journal of Environmental Sciences, 79, 256-263, https://doi.org/10.1016/j.jes.2018.11.011, 2019.

Du, M., Voliotis, A., Shao, Y., Wang, Y., Bannan, T. J., Pereira, K. L., Hamilton, J. F., Percival, C. J., Alfarra, M. R., and McFiggans, G.: Combined application of Online FIGAERO-CIMS and Offline LC-Orbitrap MS to Characterize the Chemical Composition of SOA in Smog Chamber Studies, Atmos. Meas. Tech. Discuss., 2021, 1-42, 10.5194/amt-2021-420, 2021.

Murschell, T., Fulgham, S. R., and Farmer, D. K.: Gas-phase pesticide measurement using iodide ionization time-of-flight mass spectrometry, Atmos. Meas. Tech., 10, 2117-2127, 10.5194/amt-10-2117-2017, 2017.

Stark, H., Yatavelli, R. L. N., Thompson, S. L., Kang, H., Krechmer, J. E., Kimmel, J. R., Palm, B. B., Hu, W., Hayes, P. L., Day, D. A., Campuzano-Jost, P., Canagaratna, M. R., Jayne, J. T., Worsnop, D. R., and Jimenez, J. L.: Impact of Thermal Decomposition on Thermal Desorption Instruments: Advantage of Thermogram Analysis for Quantifying Volatility Distributions of Organic Species, Environmental Science & Technology, 51, 8491-8500, 10.1021/acs.est.7b00160, 2017.

Veres, P. R., Neuman, J. A., Bertram, T. H., Assaf, E., Wolfe, G. M., Williamson, C. J., Weinzierl, B., Tilmes, S., Thompson, C. R., Thames, A. B., Schroder, J. C., Saiz-Lopez, A., Rollins, A. W., Roberts, J. M., Price, D., Peischl, J., Nault, B. A., Møller, K. H., Miller, D. O., Meinardi, S., Li, Q., Lamarque, J.-F., Kupc, A., Kjaergaard, H. G., Kinnison, D., Jimenez, J. L., Jernigan, C. M., Hornbrook, R. S., Hills, A., Dollner, M., Day, D. A., Cuevas, C. A., Campuzano-Jost, P., Burkholder, J., Bui, T. P., Brune, W. H., Brown, S. S., Brock, C. A., Bourgeois, I., Blake, D. R., Apel, E. C., and Ryerson, T. B.: Global airborne sampling reveals a previously unobserved dimethyl sulfide oxidation mechanism in the marine atmosphere, 117, 4505-4510, doi:10.1073/pnas.1919344117, 2020.

Voliotis, A., Wang, Y., Shao, Y., Du, M., Bannan, T. J., Percival, C. J., Pandis, S. N., Alfarra, M. R., and McFiggans, G.: Exploring the composition and volatility of secondary organic aerosols in mixed anthropogenic and biogenic precursor systems, Atmos. Chem. Phys., 21, 14251-14273, 10.5194/acp-21-14251-2021, 2021.

Voliotis, A., Du, M., Wang, Y., Shao, Y., Alfarra, M. R., Bannan, T. J., Hu, D., Pereira, K. L., Hamilton, J. F., Hallquist, M., Mentel, T. F., and McFiggans, G.: Chamber investigation of the formation and transformation of secondary organic aerosol in mixtures of biogenic and anthropogenic volatile organic compounds, Atmos. Chem. Phys. Discuss., 2022, 1-49, 10.5194/acp-2021-1080, 2022.

Ylisirniö, A., Buchholz, A., Mohr, C., Li, Z., Barreira, L., Lambe, A., Faiola, C., Kari, E., Yli-Juuti, T., Nizkorodov, S. A., Worsnop, D. R., Virtanen, A., and Schobesberger, S.: Composition and volatility of secondary organic aerosol (SOA) formed from oxidation of real tree emissions compared to simplified volatile organic compound (VOC) systems, Atmos. Chem. Phys., 20, 5629-5644, 10.5194/acp-20-5629-2020, 2020.

Zhang, W., and Zhang, H.: Secondary Ion Chemistry Mediated by Ozone and Acidic Organic Molecules in Iodide-Adduct Chemical Ionization Mass Spectrometry, Analytical Chemistry, 93, 8595-8602, 10.1021/acs.analchem.1c01486, 2021.

---

## Author Response (AR2)

We'd like to thank Referee 2 for their technical corrections. Below we respond to each of the points (reviewer comments in black and our responses in blue).

Minor technical details that should be corrected.

Line 306: "Despite we showed that..." please fix the grammar of this sentence.

The sentence was revised to: *Despite we showed that the volatility quantification from both approaches was challenging, the volatility derived from the partitioning calculations was more realistic and broadly comparable with the concurrent TD measurements.*

Line 382 onwards: "Note that the volatility distributions from the TD measurements are expressed as a function of the total SOA particle mass whereas the FIGAERO-CIMS as a fraction..." The part about FIGAERO-CIMS needs modification to make sense, such as "..whereas the from the FIGAERO-CIMS are shown as...".

We thank the reviewer for this suggestion. We have altered the sentence based on your suggestion that now reads as: "*Note that the volatility distributions from the TD measurements were expressed as a function of the total SOA particle mass whereas from the FIGAERO-CIMS as a function of the total signal, assuming uniform sensitivity*"

Line 615: "based on the on the" has extra repetition

Repeated words were deleted.

Line 702: "...products found in the binary ternary mixtures.." is this supposed to only have one of them mentioned, or "binary and/or ternary"?

We thank the reviewer for pointing out this typo. Indeed we were referring to binary and/or ternary so we have added the "and/or" between binary and ternary.

Line 755: "...VOC concentrations...are expected...". not "is"

We thank the reviewer for pointing out this grammatical error. We corrected this according to your suggestion.

Line 770: "...can be extrapolated globally, but to demonstrate.." instead of "rather than to". the current version is not grammatically correct.

We thank the reviewer for pointing out this grammatical error. We corrected this according to your suggestion.